# Characterizing changes in soil microbiome abundance and diversity due to different cover crop techniques

Charlotte H. Wang[1,2☯], Linda Wu[2☯], Zengyan Wang [3], Magdy S. Alabady[4], Daniel Parson[1,5], Zainab Molumo[1], Sarah C. Fankhauser [1]*

1 Oxford College of Emory University, Oxford, Georgia, United States of America, 2 Emory University, Atlanta, Georgia, United States of America, 3 Computer Sciences Department of University of Georgia, Athens, Georgia, United States of America, 4 Department of Plant Biology, University of Georgia, Athens, Georgia, United States of America, 5 Oxford College Organic Farm, Oxford, Georgia, United States of America

☯ These authors contributed equally to this work.
* sarah.fankhauser@emory.edu

**Data Availability Statement:** All relevant data are within the manuscript and its Supporting Information files. Raw sequencing data is uploaded to the NCBI BioProject repository and accessible

## Abstract

Soil-based microorganisms assume a direct and crucial role in the promotion of soil health, quality and fertility, all factors known to contribute heavily to the quality and yield of agricultural products. Cover cropping, used in both traditional and organic farming, is a particularly efficient and environmentally favorable tool for manipulating microbiome composition in agricultural soils and has had clear benefits for soil quality and crop output. Several long-term investigations have evaluated the influence of multi-mix (multiple species) cover crop treatments on soil health and microbial diversity. The present study investigated the short-term effects of a seven species multi-mix cover crop treatment on soil nutrient content and microbial diversity, compared to a single-mix cover crop treatment and control. Analysis of 16S sequencing data of isolated soil DNA revealed that the single-mix cover crop treatment decreased overall microbial abundance and diversity, whereas the control and multi-mix treatments altered the overall microbial composition in similar fluctuating trends. Furthermore, we observed significant changes in specific bacteria belonging to the phyla Acidobacteria, Actinobacteria, Planctomycetes, Proteobacteria and Verrucombicrobia for all treatments, but only the single-mix significantly decreased in abundance of the selected bacteria over time. Our findings indicate that the control and multi-mix treatments are better at maintaining overall microbial composition and diversity compared to the single-mix. Further study is required to elucidate the specific difference between the treatment effect of the multi-mix treatment and the control, given that their microbial composition changes over time were similar but they diverge into two populations of unique bacterial types by the end of this short-term study.

via the following URL: https://www.ncbi.nlm.nih.gov/bioproject/PRJNA626000/.

**Funding:** We are grateful to Oxford College of Emory University for financial support. The funders had no role in study design, data collection and analysis, decision to publish, or preparation of the manuscript.

**Competing interests:** The authors have declared that no competing interests exist.

## Introduction

Soil microbial communities are essential to the fertility of soil and success of agricultural crops, yet a substantial portion of microbial life within the soil remains uncultivated and under-explored [1]. The composition and diversity of these soil microbial communities are important indicators of the soil fertility. Within agricultural soils, microbes play essential roles in supporting plant development through processes such as nitrogen fixation, growth hormone synthesis and nutrient recycling of decomposed plant matter [2,3]. An array of farming techniques has been developed with the purpose of manipulating the microbial composition within soil in an effort to promote soil health and enhance crop quality. An approach that has demonstrated much promise in recent years is cover cropping, which is the practice of planting specific species of crop(s) to maturity then, without harvesting, mowing the cover crop back into the soil. Utilized in both conventional and organic agriculture, this technique recycles nutrients back into the soil and has been found to contribute to both pest management and the development of a beneficial biota [4]. Soils treated with cover crops have been found to have greater complexity and diversity within their microbial communities than tilled soils. Even against other sustainable techniques, including the use of organic amendments and minimum-tillage, cover cropping is comparatively more beneficial for promoting soil viability. Tillage, for example, while considered highly advantageous for soil microbiome maintenance when used conservatively, has been shown to decrease overall microbial diversity over time, whereas cover cropping has been found to consistently preserve and enhance heterogeneity [5].

A variety of classes of cover crops exist, each serving a different and specific function. Legume-based cover crops, in particular, have been found to substantially increase soil nitrogen concentration by supporting the growth of nitrogen-fixing bacteria. This particular group of cover crops has also been shown to improve overall soil quality and enhance microbial diversity in soil [6]. Of the several cover cropping practices utilized in farming, we compared the effects of single and multispecies, or multi-mix, cover crop mixtures on soil fertility and microbial abundance and diversity. It has previously been found that multi-mixtures of grass and legumes improve crop yield, microbial diversity, soil moisture, and inorganic nitrogen levels in soil when paralleled with single, double-mix, and control treatments [7]. These differences may be attributed to the apparent benefits of mixing different species of cover crops with differing actions; effective combinations of these cover crops can work to optimize each cover crops' effects on the soil. A three-year long-term study evaluating the effects of single versus multi-mix cover cropping on soybean crop quality and output observed greater crop yield in multi-mix plots than single-mix plots; the treatments' effects on microbial diversity were not assessed [6]. While other long-term studies have investigated the effects of multi-mix cover crop treatments on microbial composition and health, there is a lack of literature that explores the effects of short-term cover cropping (one year) on microbial life [7].

The present study sought to evaluate the effects of short-term cover cropping on soil microbial diversity and fertility. The study also aimed to develop a more comprehensive understanding of the impacts of combining different organic summer cover crops (multi-mix of legumes and grass as well as single-mix of buckwheat) as it relates to altering soil microbial community composition and nutrient availability, both of which directly influence crop production. We hypothesized that an increased cover crop diversity would lead to a more complex soil rhizosphere. We found that soil microbial composition and overall diversity were most altered in plots treated with the control and multi-mix cover crop treatments. While our study is based in a specific agricultural setting, our results demonstrate how specific plant communities can change the microbial ecology of the soil over a short time period.

## Materials and methods

### Experimental timeline and plot preparation

The investigation was held on the Oxford College Organic Farm in Oxford, Georgia (33.616247, -83.867004), and the soil has been identified as the Ultisol. It was certified organic in 2015 according to the USDA National Organic Program standards. A 26 x 6 m area of the Oxford Organic Farm was created in Summer 2016 by pilling neighboring soil on top of the target plot and planting Crimson Clover and grain rye in Early Fall. The cover crops were grown for three seasons until the summer of 2017, the experimental plot was designated to this study (**Fig 1A**). The plot was divided into 9 separate beds that each contained a planting strip (**Fig 1B**). The 9 subplots provided triplicates of the three treatment levels: a control, single-mix and multi-mix. Control had no cover crop treatment, single-mix consisted of the cover crop: organic non-legume buckwheat species, whereas the multi-mix was a combination of seven

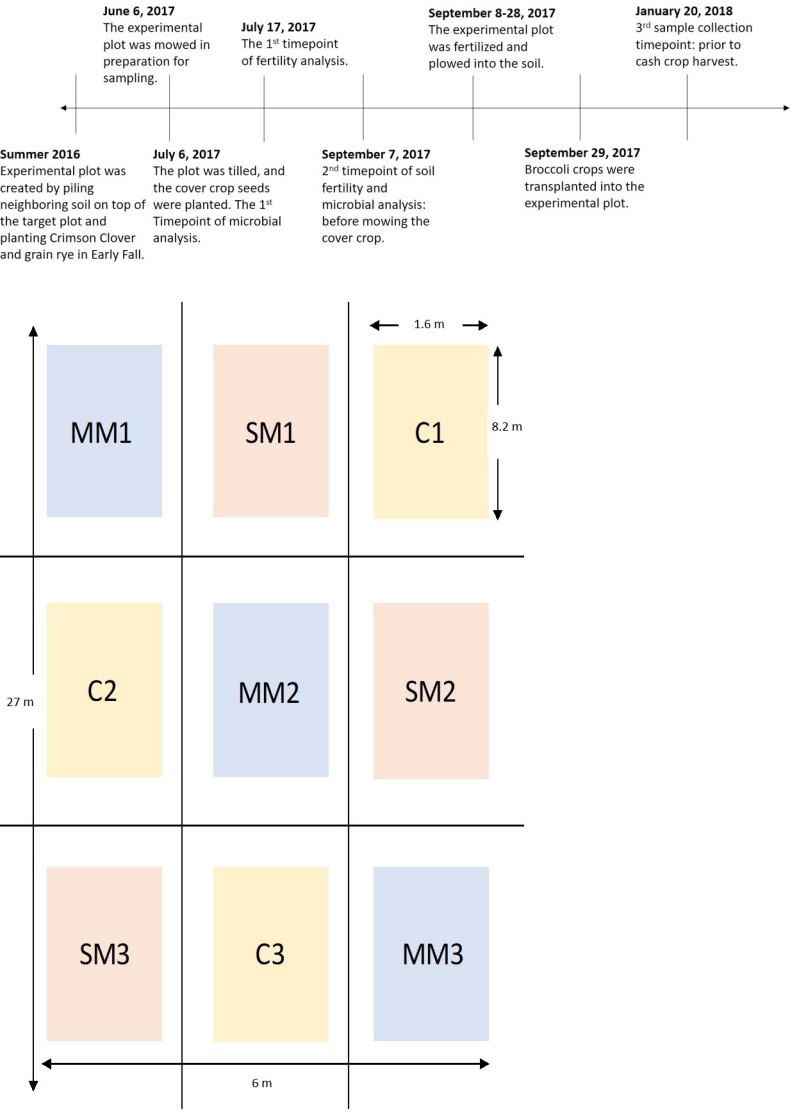

**Fig 1. A.** Timeline tracking the past plot history and key experimental time points. **B.** Outline of the experimental plots and the independent treatment groups.

summer organic cover crops: Japanese Millet, Sorghum-Sundan Grass, Pearl Millet, Soybeans, Sunnhemp, Cowpea, and Buckwheat [8]. The cover crops that were used in this study were selected in consultation with Daniel Parson, the Oxford College Organic Farmer, and based on conventional organic farming practices and economic efficiency. At the start of the study on January 6th 2017, the experimental plot was mowed in preparation for sampling. The plot was tilled, and the cover crop seeds were planted, and the initial soil analysis was conducted. A month later, the cover crops were mowed and allowed to decompose into the soil. Within 20 days, the experimental plots were fertilized and tilled. Then the cash crop, broccoli, was transplanted into the experimental plot at approximately 20 broccoli per subplot. After four months of growth and maintenance, the broccoli crop was lost due to an unexpected freeze. Data collection was initiated in June 2017 and continued until January 2018. Throughout this time period, monthly average temperatures and rainfalls were recorded and compared to previous year records (S1 Fig). A decrease in average temperature was observed compared to 2015–2016 historical month average temperatures. The summer of 2017 had an increase in total rainfall, but the winter of 2017 was drier.

## Sample collection

Microbial data collection was conducted at three key analysis time points–immediately following the planting of the cover crops (Time point 1), immediately before the mowing of the cover crops (Time point 2), and before the harvest of the broccoli crops (Time point 3) (Fig 1A). The samples were collected in triplicates for each subplot with a total of 27 samples for each time point, with the exception of time point 1 where only 1 sample per subplot was taken. There was only 1 sample per subplot taken at time point 1 since the soil had just been evenly tilled across each subplot and no further manipulation had been undertaken; thus the extra replicates of the plots were deemed unnecessary. However, we do recognize that this set up led to sample imbalances that should be taken into consideration when interpreting statistical methods used in this manuscript. Soil samples for time points 2 and 3 were collected from the rhizospheres of the plants, which were seeded initially to avoid the edges of the borders of the plot. Previous research has shown that proximity to field edges can positively impact soil nutrient content [9]. No vegetative growth was present in time point one. By the end of the experiment, there were a total of 63 soil samples used for microbial analysis. Approximately 500uL of soil was collected for each sample at using a 16 mm diameter soil probe at 100- mm depth below the soil surface. Soil samples were frozen at -80˚C for DNA extraction and further analysis.

## Soil nutrient composition

Soil tests at UGA-AESL (University of Georgia-Agricultural & Environmental Services Lab) were conducted before the cover crops are planted and before the cash crops are harvested to analyze the organic matter and nutrient availability within the soil. Additionally, cation exchange capacity was measured, which indicated the ability of organic matter to hold onto cations as well as soil fertility and nutrient retention capacity [10]. Active organic carbon was estimated using 5g of soil for each sample using $KMnO_4$ [11].

## DNA extraction and 16s rDNA sequencing

DNA was isolated from each collected soil sample using the QIAGEN DNeasyâ PowerSoilâ Kit and protocol per manufacturer's instructions.

Upon isolation and purification, the samples were sequenced at the Georgia Genomics Facility at University of Georgia where Illumina sequencing was performed using universal primers specific for the V3-V4 regions of the 16s rDNA gene.

DNA quality were assessed with a plate fluorometer, then a 16s Library was assembled. After normalizing DNA samples to 5ng/µL, 5 µL of each sample was used for primary PCR. The V3-V4 16S primers (S-D-Bact-0341-b-S-17 `CCTACGGGNGGCWGCAG` and Reverse: S-D-Bact-0785-a-A-21 `GACTACHVGGGTATCTAATCC` [12]) were diluted to 2µM. The first PCR reaction mix consisted of 2.5µL of each primer, 12.5uL of KAPA HiFi HotStart ReadyMix and 2.5uL of PCR-grade water [13]. The amplification reaction profile was: 95˚C - 3min; 15 cycles of 95˚C- 30 ec, 56˚C- 30sec, and 72˚C- 30sec; and 72˚C- 4min. A post-PCR cleanup was conducted using 0.8X AMPure beads.

The purified PCR product was resuspended in 50uL of elution buffer. Next, 5 uL of the first PCR product for each sample was used in the second PCR amplification. The reaction mix consisted of 5ul first PCR product, 5ul of each i5 and i7 Unique Illumina indexing primers, 25 uL of KAPA HiFi HotStart ReadyMix, and 10 uL of PCR-grade water. The amplification reaction profile was: 95˚C - 3min; 12 cycles of 95˚C- 30sec, 56˚C- 30sec, and 72˚C- 30sec; and 72˚C- 4min. The second PCR products were purified using 1X AMPure beads and the cleaned products were eluted in 25uL of elution buffer [13], which constitutes the final 16S sequencing libraries. The library concentrations were measured using the plate fluorometer method.

The molecular weight of the DNA was determined by running the libraries on the Fragment Analyzer instrument (Agilent) using the high sensitivity NGS kit. The NGS fragment analysis confirmed the targeted size for the V3-V4 region of the 16S amplicon. The libraries were then normalized to have equal molarity and were pooled together at equal volumes. The concentration of the pool was assessed with Qubit as well as quantitative PCR to have the most accurate reading possible. After this final quality check, the pool was ready for sequencing.

**Sequencing and post sequence analysis.** The pool was sequenced on the Miseq PE300 run following illumine standard operating procedures and recommendations [13]. Reads were de-multiplexed using the Illumina bcl2fastq. Adapters of the de-multiplexed data were trimmed by TrimGalore, the sequences were merged, and primers were removed by QIIME2 [14]. The following quality control parameters were used: reads had over 90% bases with the quality score above 30. Representative read picking was performed with the QIIME2 DADA2 denoising algorithm [15] into amplicon sequence variants (ASVs). ASVs were then assigned taxonomy using Greengenes [16], SILVA reference database (Version 132) [17].

## Statistical analysis

We used the analysis tools of the QIIME2 pipeline to perform the various statistical analyses that are mentioned in this section.

Statistical significance analysis of overall microbial composition was based on adjustments to Aitchison's norm. The Aitchison norm is a statistical analysis method used to normalize compositional data such as microbial/relative abundances. Comparing changes in relative abundances of taxa within a microbial community can present issues when looking at each unique taxon as comprising a part of the whole. Increases in abundance of a specific type of microorganism do not necessarily amount to a decrease in abundance of another or others but can be interpreted as such without first controlling for compositionality so that changes in abundance of one taxonomical unit is not correlated with any observed changes within another [18]. The Aitchison norm utilizes the isometric log ratio (ilr) transformation, which tracks changes in abundance over time and quantifies these changes as log ratio coordinates, or balances [19]. The resulting processed data can then be analyzed using multivariate

comparative methods such as ANOVA and ANOSIM. This analysis was conducted on six sample categories: 3 time points each of which has three treatments (control, single mix and multi-mix) and 3 treatments each of which has three time points (time point 1, 2, and 3). This analysis identifies whether there is significant difference between the samples in each of the categories.

The "group_significance.py" function in QIIME was also used to compare ASV abundance and to identify the statistical level of significance among samples and categories. With this function, we used two different statistical tests: ANOVA and Kruskal-Wallis (KW). The significant ASVs from both tests were cross-referenced to yield a final list of significant ASVs across different groupings ($p < 0.01$). The count of ASV indicated the abundance of each ASV, and it was normalized across samples to account for the difference in the depth of sequencing between samples, thus making it a good candidate for cross-sample comparisons. The normalized mean relative abundance for each sample was used for significance analysis. Kruskal-Wallis test was used to determine significant differences in average phylum abundance of ASVs that significantly changed over time for each treatment.

## Results

### Soil composition and fertility

**Organic carbon content.** During the six-month field trials, soil samples were collected at three distinct time points from the three different treatment plots (single-mix, multi-mix, and control) to analyze the soil organic carbon content using $KMnO_4$ extraction (**Fig 2**). A significant decrease in active carbon concentration was observed in all three treatments between time point 1 and time point 2 (2 sample t-test, $p < 0.05$). Between time point 2 and time point 3, the active carbon concentration significantly increased for all three treatments, indicating a restoration of carbon in the soil. Across treatments, the differences in active carbon concentration at respective time points were not significant (**Fig 2**).

**Cation exchange capacity.** Cation exchange capacity (CEC) was used to measure the total exchangeable cations which may be held in soil at specific pH levels. There was no significant difference between the cation exchange capacity in between the treatments at time point 3, but there was a significant increase in CEC in all three sample treatments at time point 3 in comparison to the CEC at time point 1 (two sample t-test, $p < 0.05$) (**Fig 3**).

### Overall microbial composition

The number of reads left after each step in the full amplicon workflow was recorded (**Table 1**). The full amplicon workflow included input raw reads, filtering based on quality score, denoising sequences, merging of paired-end reads and chimera filtering. The average distinct number of ASVs indicated the average total microbial life present, or the total microbial abundance in the sample. The significance of the changes in total microbial life was determined based on ANOVA and ANOSIM based on Aitchison's distances. In both the control and multi-mix treatments there was a decrease in total microbial life between time point 1 and time point 2, and an increase from time point 2 to time point 3, whereas single-mix decreased in total microbial life over time. At each respective time points, time point 2 and time point 3 showed the greatest difference between the respective treatments (**Table 2A**).

**Microbial abundance.** The average number of distinct amplicon sequences variances (ASVs) represented microbial abundance of each treatment group at respective time points. From distinct ASV numbers, we observed an increasing trend, although insignificant, in microbial abundance for the control and multi-mix treatments between time point 1 and time point 2, and a decreasing trend from time point 2 to time point 3, whereas the single mix

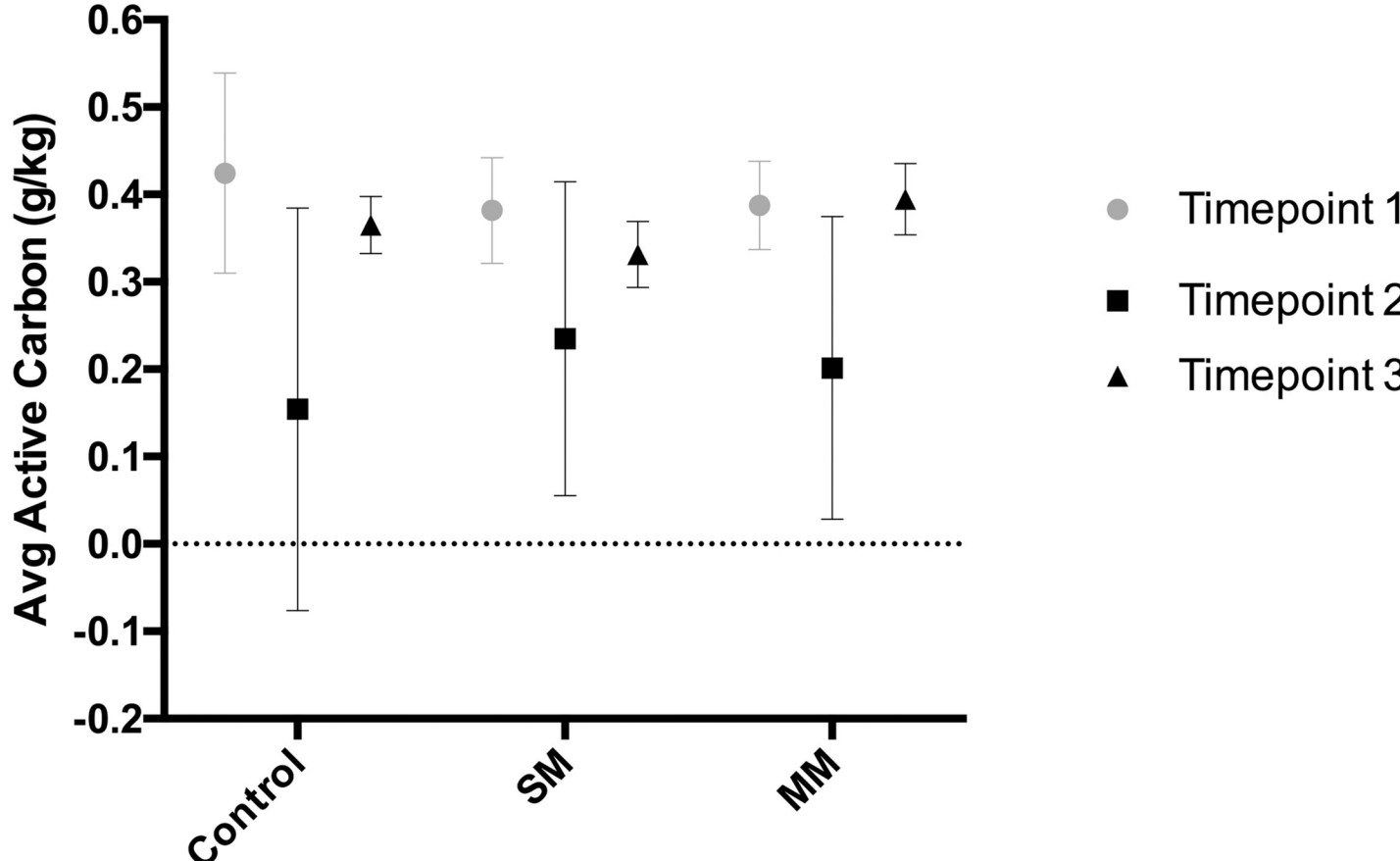

**Fig 2. Percent change in concentrations of active carbon in each subplot between time points.** Bars represent the average of 3 biological replicates. An overall 6–7% increase in active carbon content in MM and SM treatments respectively, as well as an overall 10% decrease in the control group were observed between the 1st and 2nd time point.

(nonsignificant) consistently decreased in microbial abundance over time (**Table 1**). However, according to statistical tests based on Aitchison's distances of the treatment groups, none of the changes of a treatment over time was significant (p<0.01), while we do recognize that the p value for the changes in the control over time is considerable low. The low p-value of the control (anosim_aitchison, p = 0.112; anova_aitchison, p = 0.072), compared to the higher p-values in the multi-mix and sinlg-mix treatments, indicated that our cover-cropping technique may be responsible for maintaining overall microbial abundance of the soil (**Table 2A**).

With a different analysis parameter, looking at the difference between treatments at individual time points, ANOVA and ANOSIM tests inferred a significant difference between treatments at time point 2 and time point 3 (**Table 2B**). At time point 1, the soil in every plot was untreated, and the p values at time point 1 were consistent in showing no significant difference between the soil microbial abundance. However, at time points 2 and 3, treatments started to significantly differ from each other at the same time point due to treatment effects (**Table 2B**). From distinct ASV numbers, we observed that there was indeed a significant difference between microbial abundance between the three treatments at time points 2 and 3, with the single-mix having the highest abundance at time point 2, and lowest abundance at time point 3 (**Table 1**).

**Microbial diversity.** Diversity measurements were used to compare the samples independent of their phylogenetic composition [20]. The Shannon diversity index (H) was used as a

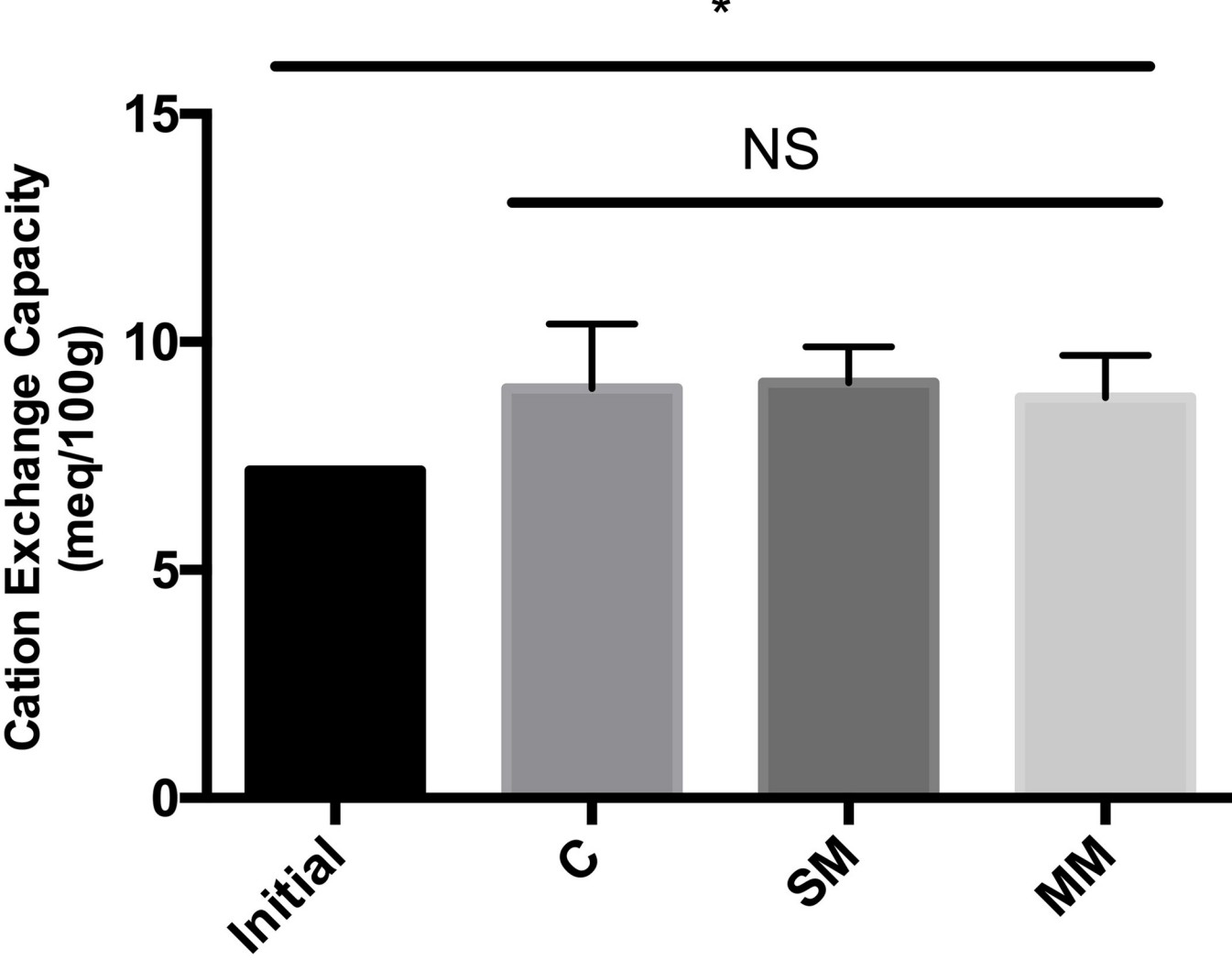

**Fig 3. Cation exchange capacity based on routine soil tests.** Samples taken from time point 1 were used as a standard to compare the effect of cover crops on soil cation exchange capacity at time point 3. A significant increase in CEC was observed in all sample treatments, indicated by the asterisk (2 sample T-test, p<0.05). Multi-mix and single-mix had similar CEC values. There was no significant difference between the treatments.

measure of general diversity, including richness and evenness [21]. Looking at the diversity of each treatment across the three time points, the Shannon diversity index was only significantly different for the control treatment between all three time points (Based on Kruskal-Wallis (p = 0.0037). Pairwise tests between the time points and Shannon indices showed that the microbial diversity for the control treatment increased between time points 1 and 2 (p = 0.014) and significantly decreased between time points 2 and 3 (p = 0.007), but the overall decrease was not significant between time points 1 and 3 (p = 0.079). The overall trend for microbial diversity in the control indicated a great fluctuation in the microbial diversity (**Table 1**). Meanwhile, the single-mix microbial diversity did not significantly change over time (Kruskal-Wallis, p = 0.017). Pairwise tests showed that the single-mix microbial diversity decreased from time point 1 to time point 2 (p = 0.052), decreased from time point 2 to time point 3 (p = 0.102). The overall decrease in diversity index from time point 1 to time point 3 (p = 0.013) inferred that the single-mix treatment had a decreasing trend in microbial diversity over time (**Table 1**). The trend of microbial diversity change in the multi-mix treatment was similar to

**Table 1. Summary statistics for each step of the full amplicon workflow and diversity index of bacterial communities.**

|  | Time Point 1 | | | Time Point 2 | | | Time Point 3 | | |
|---|---|---|---|---|---|---|---|---|---|
|  | C | SM | MM | C | SM | MM | C | SM | MM |
| Input Reads | 223909 +/- 16680 | 190305 +/- 5650 | 200699 +/- 64896 | 109408 +/- 45341 | 164517 +/- 67912 | 109856 +/- 58548 | 219351 +/- 121581 | 106797 +/- 51037 | 154522 +/- 55159 |
| Filtered Reads | 112520 +/- 11608 | 97432 +/- 4456 | 104411 +/- 33776 | 50533 +/- 23198 | 81510 +/- 35674 | 51302 +/- 28567 | 107605 +/- 58804 | 46983 +/- 27857 | 74309 +/- 27056 |
| Denoised Reads | 105244 +/- 11286 | 90931 +/- 4259 | 98519 +/- 33723 | 46997 +/- 21840 | 76499 +/- 34254 | 47590 +/- 27284 | 102213 +/- 56998 | 43450 +/- 26490 | 69467 +/- 26035 |
| Merged Reads | 83875 +/- 10130 | 72336 +/- 4124 | 81248 +/- 32897 | 37781 +/- 17771 | 62323 +/- 29205 | 37504 +/- 22643 | 86704 +/- 51071 | 34349 +/- 21881 | 56419 +/- 22797 |
| Non-chimeric Reads (Total ASVs) | 80797 +/- 9568 | 70287 +/- 4128 | 78443 +/- 31193 | 36572 +/- 17185 | 60124 +/- 27884 | 36393 +/- 21951 | 83328 +/- 48407 | 33369 +/- 21128 | 54828 +/- 22095 |
| Distinct ASVs | 2376 +/- 224 | 2183 +/- 155 | 2019 +/- 216 | 1069 +/- 483 | 1625 +/- 624 | 1154 +/- 591 | 1947 +/- 684 | 1019 +/- 558 | 1523 +/- 420 |
| Shannon | 10.24 +/- 0.14 * | 10.12 +/- 0.11 | 9.93 +/- 0.10 | 9.37 +/- 0.38 * | 9.58 +/- 0.45 | 9.53 +/- 0.39 | 9.81 +/- 0.35 * | 9.20 +/- 0.54 | 9.56 +/- 0.26 |

After the basic adapter and quality-based trimming, the number of reads left after each step in the full amplicon workflow was recorded. The denoising step included dereplication and sample inference. The drop in the number of reads during the merging step indicated paired-end sequences that could not merge, either because the sequences were too short and did not overlap, or because the ends did not align. Total ASVs included many repeated ASVs for one species, but the distinct ASV represented the count of different ASVs across all species. Shannon index was used to measure general diversity. The control and single-mix treatments altered in microbial diversity over time, whereas the multi-mix did not significantly alter microbial diversity over time.

*Significantly different from other time points of the same treatment based on ANOSIM (p<0.01)

the control—increasing between time points 1 and 2 (p = 0.079) and decreasing between time points 2 and 3(p = 0.270). The multi-mix microbial diversity, however, was not significantly different across time with respect to our alpha value of 0.01 (Kruskal-Wallis, p = 0.08). The overall change in microbial abundance between time point 1 and time point 3 also showed a decreasing trend such as the control, though the change was not significant (p = 0.052). However, the p-values were still considerably low, thus we report this p-value for consideration of the potential biological difference in microbial diversity. The trends in changes in microbial diversity were similar to the changes in microbial abundance (**Table 1**).

For the analysis of different treatments within individual time points, the difference between the diversity index of respective treatments was not significant (Kruskal-Wallis, p>0.01). However, the p values of 0.051 at time point 1, 0.10 at time point 2, and 0.070 at time point 3 demonstrated that there still may be biological differences in microbial diversity between the treatments at different time points, even at time point 1, with identical untreated soil. At time point 3, the single-mix also appeared to have the lowest Shannon diversity index (**Table 1**). This low final microbial diversity for single-mix was consistent with the overall trend that the single-mix decreases microbial diversity. Nevertheless, we do recognize that the Shannon diversity index is very sensitive to rare groups, making the analysis on microbial diversity with room for other potential inferences [21].

## Analysis of each treatment: Difference across time points

Further analysis of each treatment was based on the phylum classification of the ASV identified taxon rather than using more specific classification as the unit. Phylum analysis was utilized since ASV taxonomy assignment was inconsistent in terms of the levels of classification due to variations in reads and matches. For example, some ASVs were classified to the family level while some were classified to the genus species level of the taxonomy. In order to reduce

bias in the analysis of taxonomic assignments, the phylum level classification was most consistent across all mapped taxa, thus it was the best unit for further analysis such as changes in significantly altered bacteria over time for each treatment.

Significant differences in ASV counts were determined across the three time points for each treatment; this analysis was based on cross-references of Kruskal-Wallis (p<0.01) and ANOVA (P<0.01) tests. The control treatment plot showed the greatest number of changes, with a total of 485 ASV aligned bacterial types that differed significantly in their abundances throughout the three time points. Meanwhile, 213 bacterial types were identified to be significantly different across time points for single-mix, and 138 bacterial types for multi-mix (Table 2). Further analyses of these changes are presented below.

**Significant changes in phyla over time in control treatment.** Out of the 485 bacteria types that were significantly different in relative abundance in the control treatment, 85% of the bacterial types were classified into 14 phyla, and the remaining bacteria were unknown classifications (Acidobacteria, Actinobacteria, Armatimonadetes, Bacteroidetes, Chlorobi, Chloroflexi, Cyanobacteria, Elusimicrobia, Firmicutes, Gemmatimonadetes, Nitrospirae, Planctomycetes, Proteobacteria, Verrucombicrobia) (Table 3). Specifically, Planctomycetes and Proteobacteria were the two major phyla that comprised 45% of bacteria that significantly differed in the control treatment. The average abundance of significantly changing bacteria within each phylum, measured by ASV counts, was determined for each time point (Fig 4A). The average relative abundance of significantly changing bacteria in 11 out of the 14 phyla significantly changed from time point 1 to time point 3 (Kruskal-Wallis p<0.01), as time point 1 had the highest average abundances. This change in average abundance correlated with the overall decreasing and increasing trend in the number of distinct ASVs over time for the control treatment (Table 1). Average counts of ASVs within the phyla of Chlorobi, Elusimicrobia, and Nitrospirae did not change significantly throughout time.

**Table 2. Reported P-values of ANOVA and ANOSIM tests based on aitchison distances of microbial abundance.**

A. Reported P-value

|  | anosim_aitchison | anova_aitchison |
|---|---|---|
| Control | 0.112 | 0.072 |
| Single-Mix | 0.604 | 0.495 |
| Multi-Mix | 0.625 | 0.274 |

B. Reported P-value

|  | anosim_aitchison | anova_aitchison |
|---|---|---|
| TP1 | 0.218 | 0.165 |
| TP2 | 0.001 | 0.001 |
| TP3 | 0.007 | 0.02 |

The ANOSIM and ANOVA methods were both used to track any significant changes between each treatment group with regard to species abundance and composition over three different time points. The data were normalized using the Aitchison norm statistical analysis method. An alpha level of 0.01 was used for analysis.

A. Against this alpha level, no significant changes in abundance and/or composition were observed across any of the time points for all treatments in both the ANOSIM and ANOVA analyses.

B. For the first time point, there were no significant changes in species composition or abundance between the treatment groups. However, for both time points 2 and 3, significant changes in the microbial composition between treatment groups were supported by both the ANOSIM and ANOVA results which generated p-values below the given alpha-level value of 0.01. An exception was that at time point 3 only the ANOSIM generated a p-value lower than 0.01, but the p-value from ANOVA was also very small (p<0.05).

**Table 3. Number of bacterial types that significantly differ in relative abundance between all three time points.**

| | Total | Acidobacteria | Actinobacteria | Armatimonadetes | Bacteroidetes | Chlorobi | Chloroflexi | Cyanobacteria | Elusimicrobia | Firmicutes | Gemmatimonadetes | Nitrospirae | Planctomycetes | Proteobacteria | Verrucomicrobia |
|---|---|---|---|---|---|---|---|---|---|---|---|---|---|---|---|
| Control | 485 | 23 | 35 | 9 | 44 | 1 | 17 | 2 | 2 | 19 | 5 | 1 | 79 | 138 | 38 |
| Single-Mix | 213 | 9 | 10 | 4 | 27 | 1 | 3 | 1 | 1 | 3 | 3 | 0 | 39 | 66 | 16 |
| Multi-Mix | 138 | 8 | 7 | 3 | 4 | 0 | 11 | 0 | 1 | 17 | 4 | 1 | 27 | 30 | 9 |

Significant difference was determined based on cross references from Kruskal-Wallis (p<0.01) and ANOVA (P<0.01) tests on average ASV counts of different bacterial types. The number of bacterial types that significantly differed in relative abundance based on ASV counts for the control was more than twice the number of bacterial types that significantly differed in relative abundance for the single and multi-mixed treatments.

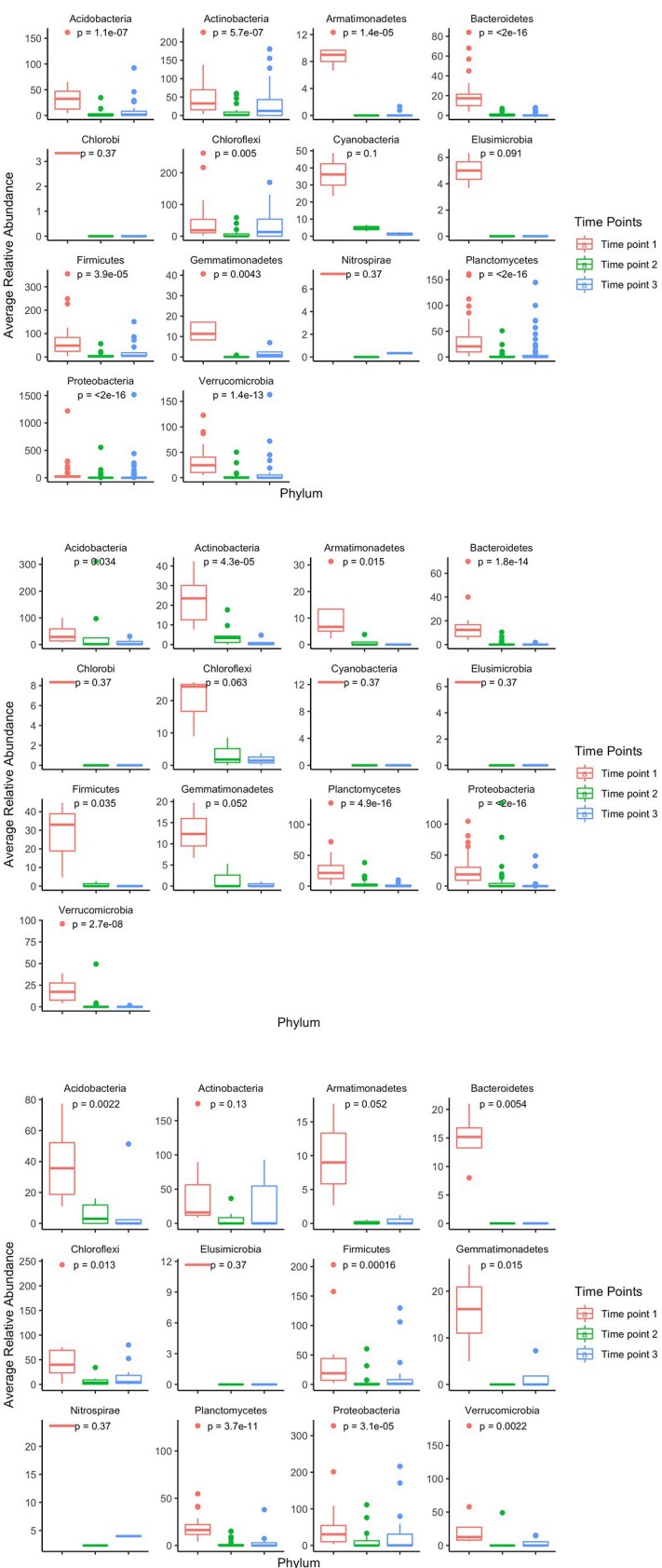

**Fig 4. Change in average relative abundance of significantly different ASVs within distinct phyla classifications at all three time points for the three different treatments. A.** For the control treatment, the change in average relative abundance of significantly different bacteria taxa in their relative phyla classifications generally decreased throughout consecutive time points. Out of the 14 classified phyla, 11 demonstrated a significant change in average phyla relative abundance from time point 1 to time point 3. **B.** For the single-mix treatment, the change in average relative abundance of significantly different bacteria taxa in their relative phyla classifications generally decreased throughout consecutive time points. Out of the 8 classified phyla, 4 demonstrated a significant decrease in average phyla relative abundance between time point 1 and time point 3. **C.** For the multi-mix treatment, the change in average relative abundance of significantly different bacteria taxa in their relative phyla classifications generally increased throughout consecutive time points. Out of the 10 classified phyla, 5 demonstrated a significant change in average phyla relative abundance from time point 1 to time point 3.

**Significant changes in phyla over time in single-mix treatment.** Of the 213 bacteria types that were significantly different in relative abundance across time in the single-mix treatment, 86% were classified into 13 different phyla (Acidobacteria, Actinobacteria, Armatimonadetes, Bacteroidetes, Chlorobi, Chloroflexi, Cyanobacteria, Elusimicrobia, Firmicutes, Gemmatimonadetes, Planctomycetes, Proteobacteria, Verrucombicrobia), with the remaining taxa having unknown classifications (**Table 3**). Unlike the control treatment, the average abundance for significantly changing bacteria within 5 phyla significantly decreased from time point 1 to time point 3 (Kruskal-Wallis, p<0.01), similar to the decreasing trend of overall microbial abundance and diversity for the single-mix treatment (**Fig 4B, Table 1**).

**Significant changes in phyla over time in multi-mix treatment.** Of the 138 bacterial types that were significantly different in relative abundance (ASV counts) across time in the multi-mix treatment, 88% were classified into 12 phyla (Acidobacteria, Actinobacteria, Armatimonadetes, Bacteroidetes, Chloroflexi, Elusimicrobia, Firmicutes, Gemmatimonadetes, Nitrospirae, Planctomycetes, Proteobacteria, Verrucombicrobia) (**Table 3**). The average relative abundance for 6 significantly changing bacterial taxa classified phyla significantly changed between time point 1 and time point 3 (Kruskal-Wallis p<0.01), as time point 3 had the greatest average abundances (**Fig 4C**). The average relative abundance of OTUs within the phyla of Actinobacteria, Armatimonadetes, Chloroflexi, Elusimicrobia, Gemmatimonadetes, and Nitrospirae did not change significantly throughout time, however, these phyla also had fewer significantly changing bacteria taxa in each of these phyla.

Notably, the total ASV count changed from time point 1 to time point 3 for the multi-mix treatment in a similar trend as the control, which is consistent with the rest of the total microbial composition analysis (**Table 1**).

Overall, the microbial composition trends for all three treatments over time were consistent with the phylum grouped analysis of specific bacteria taxa what significantly changed over time. Notably, the most predominant changing bacteria phyla classification such as Bacteroidetes, Planctomycetes, Proteobacteria, and Verrucombicrobia indicated significant changes in average ASV abundance. These phyla are important to consider in discerning the trends and effects of our cover cropping treatments in the future.

## Analysis of unique bacterial types across various time points

While analyzing overall microbial composition changes of respective treatments, we recognized that the overall trend was that single-mix had a decrease in overall microbial composition whereas control and multi-mix treatments alter the microbiome in similar trends. In order to elucidate the difference between the control and multi-mix treatment, even their difference with the single-mix treatment, we analyzed populations of bacteria that were unique to each treatment group at the respective time points (**Fig 5**). In order to find the truly unique bacterial species that were present in one treatment but not the other, we were very strict in

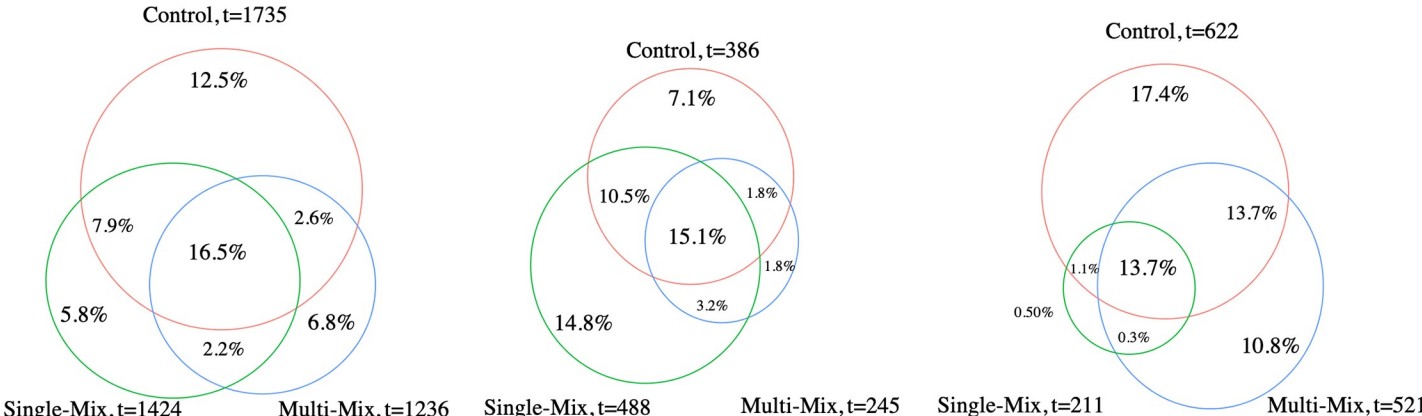

**Fig 5. Change in unique bacterial types for each treatment at different time points.** All identified bacterial taxa were filtered by their respective ASV counts for individual samples. **A**. At time point 1, there were 1735 bacterial types from the control, 1424 from the single-mix and 1236 bacterial types from the multi-mix that had more than 1/3 of the replicates with nonzero ASV counts. Of all bacterial types analyzed, 16.5% had common bacterial type identifier codes, indicating the same type of bacteria, while unique bacterial types were identified for each treatment group. **B**. At time point 2, there were 386 bacterial types from the control, 488 from the single-mix and 245 bacterial types from the multi-mix that had more than 1/3 of the replicates with nonzero ASV counts. Of all bacterial types analyzed, 15.1% had common bacterial type identifier codes, indicating the same type of bacteria, while unique bacterial types were identified for each treatment group. **C**. At time point 3, there were 622 bacterial types from the control, 211 from the single-mix and 521 bacterial types from the multi-mix that had more than 1/3 of the replicates with nonzero ASV counts. Of all bacterial types analyzed, 13.7% had common bacterial type identifier codes, indicating the same type of bacteria, while unique bacterial types were identified for each treatment group.

the filtering criteria) and did not allow for a filtering analysis based on average ASV counts as some samples may have a high ASV count in only one of the three technical replicates, which was still filtered out in our unique bacteria analysis in order to eliminate all potential uncertainties or sequencing errors. The taxa filtering step eliminated all bacterial taxa that had more than 1/3 samples with zero ASV counts. After the rigorous filtering step, only 9.6% of bacteria from time point 2, 10.2% of bacteria from time point 3 met the filtering criteria and were included in the comparison for unique taxa at respective time points (rowSums(n == 0)>3). However, due to the fewer replicates at time point 1, the filtering was not as selective for time point 1 and 54% of all bacterial types were selected for analysis of unique bacterial taxa (rowSums(n == 0)>1). Distinct identifier codes assigned during taxonomy assignment were used as a unit of measurement of taxa. Thus, the resulted list of unique bacteria may have similar taxon classifications, but were unique in their ASV mapping.

At time point 2, both multi-mix and control decreased in the percentage of unique bacterial taxa, while the single mix increased in its proportion of taxa that were unique. At time point 3, the reverse occurred, while the single-mix had only 7 taxa that were unique. For the unique bacterial types in the single-mix, none of them were able to be classified beyond the family level. This change in unique taxa distribution from time point 1 to time point 3 confer with the trend that the single-mix decreased in overall microbial composition, while the control and multi-mix increased in unique bacterial taxa by time point 3, and the divergence between the bacterial taxa at time point 3 was analyzed further to draw out differences between the effect of control and multi-mix treatment at the bacterial genus level.

There were 236 bacterial types unique to the control treatment at time point 3, notably, some examples include: *Actinomadura*, *Methylocapsa*, *Telmatonacter*, *Candidatus Xiphinematobacter*, *Actinoallomurus*. Meanwhile, there were 147 bacterial types unique to the multi-mix treatment, but all the genus level classifications of the unique bacterial types in the multi-mix were the same as the control, despite having different identifier codes. Thus, different bacterial types were unique to the control and multi-mix treatments due to their differences in

amplicon mapping, but they had similar taxonomical classifications. More inferences based on the unique bacterial genera identified in each treatment helped elucidate the difference between the multi-mix and control treatments.

## Discussion

Our investigation aimed to understand the impact of various organic cover crop strategies on the microbial composition within the soil as well as the resulting impact of soil fertility. We hypothesized that the higher cover crop diversity would result in a more complex soil rhizosphere, diverse microbial community composition, and higher crop production. Overall, our results supported our hypothesis because the multi-mix treatment had greater diversity relative to the single-mix conditions at the end of the field study. Our preliminary analysis also suggests that the multi-mix treatment promoted the abundance of certain bacteria that could potentially provide benefits for plant growth. Thus, our results show that multi-mix is likely a more favorable option compared to single-mix for farmers if the focus is to maintain microbial diversity.

### Soil chemical composition

Based on our KMnO4 extraction, we found that that the active carbon content of the soil didn't significantly differ across treatments but rather changed across time (**Fig 2**). Between time points 1 and 2, the carbon content decreased, but following time point 2, and the planting of the broccoli crop, the carbon content increased. The consistency between treatments is similar to what was observed in a long-term study looking at the effects of legume, non-legume cover crops, and control treatments on soil organic carbon content [22]. Similarly, that study showed few differences between the control, non-legume cover crop, and low-legume cover crop treatment. Rather, carbon levels are likely to shift during the cycling of carbon through the cover crop and broccoli growth periods. Based on previous reports, organic carbon levels tend to decrease throughout the cultivation period due to the disturbances to the soil, which exposes the organic matter and allows for its decomposition [23]. This observation may explain why our carbon content decreased from time point 1 to time point 2 (**Fig 2**). In addition, broccoli is a C3 plant, and previous studies on legume-based cropping have shown that C3 plants provide significant soil organic carbon increases through plant residues [24, 25]. This could explain the carbon content increase between time points 2 and 3 when the broccoli growth took place (**Fig 2**).

Similar to our carbon data, our results on Cation Exchange Capacity (CEC) show that they did not differ based on the cover crop treatment, but increased significantly over time (**Fig 3**). CEC values are important determinants of soil chemical quality, particularly the retention of major nutrient cations Ca, Mg and K, and immobilization of potentially toxic cations Al and Mn. It can be indicators of soil health as well as soil capacity to absorb nutrients, pesticides, and other chemicals [26]. Our data suggest that cover crop treatments do not significantly affect the soil chemical absorption capacity.

Together, based on our results, we did not find an association between the chemical composition and the cover crop treatments. Instead, we found that time played a larger role in influencing the active carbon content and CEC of the soil. Previous literature has shown that soil organic carbon content can directly affect the soil microbial community composition [27] and organic carbon content has been found to be a limiting factor in soil bacterial development [28]. Interestingly, although we observed an increase in the organic carbon content in time point 3 across all treatments, it did not correlate to consistent changes in microbial diversity. Based on our data, there are other factors beyond carbon content and CEC that contribute to

the changes in microbial diversity. While the short timeframe of study could explain the lack of association between treatment and chemical composition, others have reported differences across treatments within as few as 60 days [29]. This suggests that even if we extended the timeframe for our study, we may not see substantial changes in CEC and carbon content across the different treatments. Thus, our data demonstrate that the cover crop treatments tested do not substantially impact the carbon content and CEC, and that other measurements, such as microbial diversity, may be more valuable in examining differences in soil profiles.

## Diversity measure

Based on the Shannon diversity index, the single-mix treatment demonstrated consistent decreasing microbial diversity over time while the control and multi-mix treatment groups showed a similar trend of decreasing then increasing in diversity over time (**Table 1**). In fact, by time point 3, single-mix showed the lowest diversity index, suggesting that this treatment likely contributed to decreasing the microbial diversity of the soil over time (**Table 1**). Compared to single-mix, multi-mix and control mix decreased between time point 1 and 2, and increased between time point 2 and 3. In the past, other studies have also observed a temporary decrease in microbial diversity when plant diversity is at its peak, and this supports the same observation in our results [30]. Subsequently, the wide variety of plants in the control and multi-mix treatments likely contributed to the increasing diversity between time points 2 and 3 due to the influence of decomposing plant litter on the activity and community structure of soil microbial communities [31]. This effect has been shown across many types of ecosystems including grasslands, croplands, and natural forests [32, 33]. All together, these results show that limitations on plant diversity can negatively impact microbial diversity as well. Thus, in a practical sense, when determining which treatment may be better for increasing microbial diversity over time, we conclude that multi-mix is better than single-mix for the purpose of increasing potential beneficial microbial diversity. Further analysis remains to be conducted on the specific groups of microbes that are unique to the treatments.

It is also worth noting that the Shannon index is very sensitive to rare groups, meaning that while some groups may exist in very low abundance, they still represent part of the overall diversity in the samples [21]. However, when we look at the ANOSIM and ANOVA analysis of Aitchison distances of overall microbial composition, we find that at time point 2 and 3, the treatments differ significantly in microbial abundance. Notably, single-mix showed the highest abundance in time point 2 and the lowest abundance in time point 3 (Table 1). This suggests that within the process of decomposing the cover crops and growing the broccoli crop, the microbial abundance of multi-mix and control treatments increased more than that of single-mix. This may be due to the increased diversity of plant litter that the soil was exposed during the process of decomposition, thus allowing a greater abundance of microbial growth in the treatments with higher plant diversity. This observation is consistent with other studies that observed an increase in microbial biomass with increased plant diversity [31].

Overall, our data, along with the literature, support the conclusion that the increased diversity of cover crops promotes greater microbial diversity of the soil. And together with the soil chemical analysis, this change in microbial diversity is independent of carbon content and CEC. This suggests that other environmental factors beyond carbon content and CEC, such as pH, may be a larger determining factor in the soil microbial diversity [34].

## Major phyla and differences across treatments

Within each treatment, we identified bacterial types with significantly changing abundance (ASV counts) and classified them into their respective phyla and compared the changes at the

phyla level across the time points. Notably, 485 bacterial types significantly changed in relative abundance in the control treatment, while only 213 and 138 bacterial types significantly changed in relative abundance in the single-mix and multi-mix, respectively. In the control treatment, the soil was more exposed to environmental and weather changes due to the lack of substantial vegetation covering the soil which may have contributed to the higher fluctuations of microbial abundances [35, 36].

While the multi-mix treatment showed fewer changing ASVs compared to the control, it had more compared to the single-mix. Additionally, while we observed significant decreases in the phyla-level abundances of certain bacterial types over time in the single-mix treatment, the multi-mix showed a decrease then an increase in the abundance of some phyla-level bacterial types (**Fig 4B and 4C**). Analysis of the significantly changing bacterial types within the multi-mix treatment suggests that the multi-mix cover cropping process may play a role in promoting the growth of bacteria that could potentially provide benefits like rhizoremediation and moisture retention [37, 38, 39, 40].

Within all of the treatments, there were significant fluctuations in certain types of bacteria with known beneficial properties (**Fig 4**). For example, abundance of Proteobacteria, which includes a wide variety of both pathogens and nitrogen-fixing bacteria, increased in the multi-mix and control treatments between time points 2 and 3 [37]. Bacteroidetes, another phylum that significantly changed in abundance across time in all treatments, are known to specialize in the organic matter degradation and is considered a biological indicator of agricultural soil usage [38]. Planctomycetes also altered significantly across time among all treatments. It is known for anaerobic ammonium oxidation and it is associated with low nitrogen fertilizers [39]. Even though some phyla were observed to decrease in abundance in treatments, some unique genera increased in abundance in the multi-mix and control treatments by time point 3, suggesting that these treatments could promote the growth of microbes to that could potentially facilitate agricultural development (**Fig 4**).

## Unique bacterial types

Concurrent with our other data, we observed that the number of unique bacteria for single-mix decreased over time. In comparison, the control and multi-mix treatment first decreased then increased in unique bacterial taxa by time point 3 (**Fig 5**). Based on our analysis of unique bacteria types at time point 3 in the treatments, we found many potentially beneficial types existing in both control and multi-mix. Many bacterial types have been previously isolated from soil samples and identified for their potential beneficial roles. For example, *Actinomadura* is a known producer of antiviral and antibiotic compounds, *Methylocapsa* oxidizes atmospheric methane aerobically and assimilates carbon from both methane and carbon dioxide, and *Telmatonacter* is also a potential beneficial bacterium found in many soil communities [40, 41, 42]. *Candidatus Xiphinematobacter* serves important functions in the degradation of complex organic compounds like cellulose and starch and *Actinoallomurus* were suggested as plant-growth-promoting agents in the past [43, 44]. However, the limits of detection confined our ability to identify the bacterial types much beyond the genus level, thus, we still cannot exclude the possibility that the multi-mix treatment could contribute to the increase in potentially harmful bacteria. We now have a repository list of bacteria to analyze in the future as our understanding of the microbiome improves. By then, we may develop a clearer picture of the positive and/or negative roles that these bacterial types can play in the agricultural process.

Our data are even more promising given that we have observed significant changes in relatively small plots. Taking into account the edge effect on changing soil nutrient conditions, we considered the possibility that the changes in microbial diversity could have been influenced

by the changing soil conditions near the edge by neighboring vegetation. In several studies in edge effect, field edges have been associated with increased organic C content [8, 45], however, in our samples, we did not find an increase in the total ASV abundance in our treatments that would likely be associated with this effect [27]. While we did observe an increase in organic C content over time, this could be influenced by the edge effect or the cycling of cover crop and broccoli growth. Although we are unable to determine the predominant influence for the phenomenon and it remains a possibility that the edge effect impacted carbon and other nutrient levels, ultimately these soil fertility changes were independent of the microbial diversity changes that we observed.

Given our observations, and considering the limitations to our analyses, our results suggest that the multi-mix and control treatment selects for potentially agriculturally beneficial bacteria while also maintaining overall microbial composition compared to a single-mix treatment. Our data contributes to the growing body of knowledge of how specific agricultural processes can significantly alter the soil microbiome, which may have potential impacts for the health and success of crops. Furthermore, our results provide a practical farming application viable to both conventional and organic farmers. Future investigations include analyzing the specific roles of the bacteria and the effect of these selected bacteria on metabolic processes such as nitrogen fixation.

## Supporting information

**S1 Fig.** Average monthly temperature **(A)** and average total rainfall **(B)** in comparison to data from past two years. A decrease in average temperature was observed. The summer of 2017 had an increase in total rainfall, but the winter of 2017 was drier.
(TIF)

**S1 Table. Cover crop characteristics and statistics.**
(TIF)

**S2 Table. Aitchison distance tables of normalized distances between different time points of all the treatments.**
(TIF)

**S3 Table. Aitchison distance tables of normalized distances between different treatments within each individual time point.**
(TIF)

**S1 File. Analysis of ASV abundance of all identified bacterial types in the control treatment.** ANOVA (S1 File -> anova_ocs.txt) ad Kruskal-Wallis (S1 File -> kw_ocs.txt) tests were performed for the control treatment to find bacterial types that significantly differed in ASV counts across time points. The list of significantly changing bacterial types was included (S1 File -> control_sig.csv) as well as summary statistics (average and standard deviation of abundance) of significantly changing bacteria categorized by their phyla classifications (S1 File -> control_significant_taxa_stats.csv).
(ZIP)

**S2 File. The alpha rarefication analysis for control.** The alpha rarefication analysis was used to determine if the richness of samples has been fully observed or sequenced. The alpha diversity rarefication plot was included (S2 File -> alpha_rarefaction -> rarefaction.qzv) to illustrate different rarefaction measurements such as Shannon and number of OTUs. It showed that richness of all treatments was achieved with sequence depth of about 10000 reads. All treatments received sequencing depth that was much more 70,000. This suggested that the

analysis discovered all microbes in these samples. Statistical tests of Shannon indices were included (S2 File -> Shannon_group_significance.qzv). PC analysis of reference sequence categorized by samples was included in the output qzv file (S2 File -> beta_rarefaction -> weighted_unifrac.qzv) using weighted UniFrac distance.
(ZIP)

**S3 File. Analysis of OTU abundance of all identified OTUs in the single-mix treatment.** ANOVA (S3 File -> anova_ocs.txt) ad Kruskal-Wallis (S3 File -> kw_ocs.txt) tests were performed for the single-mix treatment to find bacterial types that significantly differed in ASV counts across time points. The list of significantly changing bacterial types was included (S3 File -> singleMix_sig.csv) as well as summary statistics (average and standard deviation of abundance) of significantly changing bacteria categorized by their phyla classifications (S3 File -> singleMix_significant_taxa_stats.csv).
(ZIP)

**S4 File. The alpha rarefication analysis for single-mix.** The alpha rarefaction analysis was used to determine if the richness of samples has been fully observed or sequenced. The alpha diversity rarefaction plot was included (S4 File -> alpha_rarefaction -> rarefaction.qzv) to illustrate different rarefaction measurements such as Shannon and number of OTUs. It showed that richness of all treatments was achieved with sequence depth of about 10000 reads. All treatments received sequencing depth that was much more 70,000. This suggested that the analysis discovered all microbes in these samples. Statistical tests of Shannon indices were included (S4 File -> Shannon_group_significance.qzv). PC analysis of reference sequence categorized by samples was included in the output qzv file (S4 File -> beta_rarefaction -> weighted_unifrac.qzv) using weighted UniFrac distance.
(ZIP)

**S5 File. Analysis of OTU abundance of all identified OTUs in the multi-mix treatment.** ANOVA (S5 File -> anova_ocs.txt) ad Kruskal-Wallis (S5 File -> kw_ocs.txt) tests were performed for the multi-mix treatment to find bacterial types that significantly differed in ASV counts across time points. The list of significantly changing bacterial types was included (S5 File -> multiMix_sig.csv) as well as summary statistics (average and standard deviation of abundance) of significantly changing bacteria categorized by their phyla classifications (S5 File -> multiMix_significant_taxa_stats.csv).
(ZIP)

**S6 File. The alpha rarefication analysis for multi-mix.** The alpha rarefication analysis was used to determine if the richness of samples has been fully observed or sequenced. The alpha diversity rarefaction plot was included (S6 File -> alpha_rarefaction -> rarefaction.qzv) to illustrate different rarefaction measurements such as Shannon and number of OTUs. It showed that richness of all treatments was achieved with sequence depth of about 10000 reads. All treatments received sequencing depth that was much more 70,000. This suggested that the analysis discovered all microbes in these samples. Statistical tests of Shannon indices were included (S6 File -> Shannon_group_significance.qzv). PC analysis of reference sequence categorized by samples was included in the output qzv file (S6 File -> beta_rarefaction -> weighted_unifrac.qzv) using weighted UniFrac distance.
(ZIP)

**S7 File. List of unique bacterial types for each time point.** For every time point, there was a full list of all the bacterial types that were unique to each treatment summarized in a csv file (ex. "S7 File -> tp1 -> C_unique.csv" was the list of unique bacterial types for the control

treatment at time point 1) including the identifier code, taxonomy code, ASV count for each bacterial type. Relatively unique populations were also listed for future references.
(ZIP)

## Acknowledgments

We are grateful to Dr. Emily McLean for assistance with statistical analysis, and Dr. Latonia Taliaferro-Smith for her mentorship.

## Author Contributions

**Conceptualization:** Linda Wu, Daniel Parson, Sarah C. Fankhauser.

**Data curation:** Zengyan Wang, Magdy S. Alabady, Sarah C. Fankhauser.

**Formal analysis:** Charlotte H. Wang, Linda Wu, Zengyan Wang, Magdy S. Alabady, Sarah C. Fankhauser.

**Funding acquisition:** Sarah C. Fankhauser.

**Investigation:** Charlotte H. Wang, Sarah C. Fankhauser.

**Project administration:** Sarah C. Fankhauser.

**Resources:** Sarah C. Fankhauser.

**Supervision:** Sarah C. Fankhauser.

**Writing – original draft:** Charlotte H. Wang, Linda Wu.

**Writing – review & editing:** Charlotte H. Wang, Linda Wu, Zainab Molumo, Sarah C. Fankhauser.

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
