## [Decision Letter · Decision Letter 0]

7 Nov 2019

PONE-D-19-19916

Characterizing changes in soil microbiome abundance and diversity due to different cover crop techniques

PLOS ONE

Dear Dr Fankhauser,

Thank you for submitting your manuscript to PLOS ONE. After careful consideration, we feel that it has merit but does not fully meet PLOS ONE’s publication criteria as it currently stands. Therefore, we invite you to submit a revised version of the manuscript that addresses the points raised during the review process.

We would appreciate receiving your revised manuscript by Dec 22 2019 11:59PM. To enhance the reproducibility of your results, we recommend that if applicable you deposit your laboratory protocols in protocols.io, where a protocol can be assigned its own identifier (DOI) such that it can be cited independently in the future. For instructions see: http://journals.plos.org/plosone/s/submission-guidelines#loc-laboratory-protocols

We look forward to receiving your revised manuscript.

Kind regards,

Primo Proietti

Academic Editor

PLOS ONE

Journal Requirements:

1. Please include captions for your Supporting Information files at the end of your manuscript, and update any in-text citations to match accordingly. Please see our Supporting Information guidelines for more information: http://journals.plos.org/plosone/s/supporting-information.

Additional Editor Comments (if provided):

Your manuscript has now been reviewed by two reviewers. You will see that, while they find your work of interest, they have raised points that need to be addressed by a major revision.

We therefore invite you to revise and resubmit your manuscript, taking carrefully into account the points raised and the changes requested by the reviewers.

Reviewers' comments:

Reviewer's Responses to Questions

**Comments to the Author**

1. Is the manuscript technically sound, and do the data support the conclusions?

Reviewer #1: Yes

Reviewer #2: Partly

2. Has the statistical analysis been performed appropriately and rigorously? 

Reviewer #1: Yes

Reviewer #2: No

3. Have the authors made all data underlying the findings in their manuscript fully available?

Reviewer #1: Yes

Reviewer #2: No

4. Is the manuscript presented in an intelligible fashion and written in standard English?

Reviewer #1: Yes

Reviewer #2: Yes

5. Review Comments to the Author

Reviewer #1: Dear Editor, the paper “Characterizing changes in soil microbiome abundance and diversity due to differentcover crop techniques” is a study of the microbial composition of soils treated with single and multi-mix covercropping throughout a one year activity in organic agriculture regimen.

The article structure is sound, being organized around the hypothesis that multi-mix cover crop has an advantage on simpler cover crop choices. The statistics and the microbial methodology are correct. There are some major and minor points to improve, as listed below.

Major points

1. I have some doubts on the meaning and effect of these tiny experimental plots and if (even in the center of each plot) the border effect can cause problems of any type. In any case a discussion on this point seems necessary as well as the position of the sampling within the plot

2. The OUTs are described only by stating that the standard QIME setting was used. In my experience, sometimes, QIME is indeed set up to define ASV rather than OTUs. The great number of OTUs makes me think that indeed they could be ASV groups. In any case, for clarity sake, the authors should state what was the similarity threshold chosen.

3. As an addition, I would also encourage the authors to briefly describe the variability within the OTUs in order to understand whether they are largely clonal or real OTUs with a species like structure.

4. The authors claim that the changes of the mixes are induced by variations in the rhizosphere, which is quite likely. Now the point is why did not the authors sample the soil among the roots separately from the rest if they wanted to propose and support this hypothesis? Is there any other indirect evidence in favor of this very likely hypothesis, beyond the likelihood itself?

5. The authors state at the beginning that this is one of the few papers on the one-year effect of cover cropping but fail to draw any conclusion about it in the course of the paper. For instance, the relative insensitivity or carbon content to the treatments, can be an effect of the short experimental time?

6. In general the paper should be a bit more condensed to ease its reading. The discussion is partly an enlargement of the result section, rather than a real discussion.

Minor points

1. Line 165- it is unclear how can the authors sample 500 ul, i.e half cubic centimeter, from a sampling cylinder with a 16 mm diameter and 100 mm depth. I would conclude that it is 64*pi *100 = 20.096 cubic millimeters i.e. ca 20 mL. In any case the author are encouraged to use Standard measures and therefore mm for lengths an mL or uL (no ul ) for volumes.

2. Line 199 – Fragment Analyzer. Is it the Agilent tool? If so state it, please, along with the experimental settings.

3. Lines 208-209 – rephrase: it is hard to understand

4. Line 218 – is it CRAN R anosim function? If so state it and acknowledge it with a proper citation. Moreover state its settings.

5. ANOVA and KW were carried out with or what?

6. Table 1, please separate thousands with the standard “,” in order to make the figures more readable.

7. Line 564 – sentence without the verb, apparently.

8. Iconography taxonomic names not in italic.

Reviewer #2: Review

1. I don't believe the authors have submitted raw sequencing data to a public repository, such as NCBI SRA. While it might not be a strict requirement, doing so greatly improves the reproducibility of the study and enables other researchers to improve on the analyses carried out by the authors.

2. (Lines 162-163) The authors should explain, why there is only one replicate per subplot for the first time-point. Considering that the authors have used ANOVA and ANOSIM, both of which are quite sensitive to sample imbalances, the authors should emphasise this issue.

3. (Lines 210-213) OTU picking is an extremely noisy process that is extremely prone to overestimating absolute diversity (i.e. the number of observed OTUs) [1,2,3,4]. Moreover, UCLUST is one of the least accurate heuristic sequence clustering algorithms [3]. The authors must apply denoising to reconstruct exact amplicon sequence variations using DADA2 [4] or Deblur.

4. (Line 218) ANOSIM operates on dissimilarity matrices: the authors must specify the (dis)similarity functions they've used to generate these matrices.

5. (Lines 219-221, 229) ANOVA, Kruskal-Wallis and t-test cannot be applied to raw compositional data, such as sequencing data and chemical compositions [5,6]: the authors must carry out statistical analyses designed for compositional data.

6. (Lines 271, 283-285) Absolute diversity analysis is neither reliable (due to aforementioned issues with OTU picking), not subcompositionally coherent. The same critique applies to Chao and Shannon indices. If the authors do want to conduct alpha-diversity analysis, they can use the Aitchison's norm. Otherwise, alpha-diversity analyses must be removed.

References:

1. Edgar, R. (2017). Accuracy of microbial community diversity estimated by closed- and open-reference OTUs PeerJ 5(), e3889. https://dx.doi.org/10.7717/peerj.3889

2. Edgar, R. (2018). Updating the 97% identity threshold for 16S ribosomal RNA OTUs Bioinformatics 34(14), 2371-2375. https://dx.doi.org/10.1093/bioinformatics/bty113

3. Chen, W., Zhang, C., Cheng, Y., Zhang, S., Zhao, H. (2013). A Comparison of Methods for Clustering 16S rRNA Sequences into OTUs PLoS ONE 8(8), e70837. https://dx.doi.org/10.1371/journal.pone.0070837

4. Callahan, B., McMurdie, P., Rosen, M., Han, A., Johnson, A., Holmes, S. (2016). DADA2: High-resolution sample inference from Illumina amplicon data Nature Methods 13(7)https://dx.doi.org/10.1038/nmeth.3869

5. Gloor, G., Macklaim, J., Pawlowsky-Glahn, V., Egozcue, J. (2017). Microbiome Datasets Are Compositional: And This Is Not Optional Frontiers in Microbiology 8(), 2224. https://dx.doi.org/10.3389/fmicb.2017.02224

6. Quinn, T., Erb, I., Richardson, M., Crowley, T. (2018). Understanding sequencing data as compositions: an outlook and review Bioinformatics 34(16), 2870-2878. https://dx.doi.org/10.1093/bioinformatics/bty175

6. PLOS authors have the option to publish the peer review history of their article (what does this mean?). If published, this will include your full peer review and any attached files.

Reviewer #1: Yes: G. Cardinali

Reviewer #2: No

---

## [Author Response · Author response to Decision Letter 0]

20 Dec 2019

In several instances, we refer to figures and data generated in response to reviewer comments. These figures can be found in the response letter that we uploaded.

 We have reviewed the style requirements and ensured that files are named appropriately.

Please include captions for your Supporting Information files at the end of your manuscript, and update any in-text citations to match accordingly. Please see our Supporting Information guidelines for more information: http://journals.plos.org/plosone/s/supporting-information.

We have included captions for all supporting information files at the end of the manuscript as requested.   

Have the authors made all data underlying the findings in their manuscript fully available?  The PLOS Data policy requires authors to make all data underlying the findings described in their manuscript fully available without restriction, with rare exception (please refer to the Data Availability Statement in the manuscript PDF file). The data should be provided as part of the manuscript or its supporting information, or deposited to a public repository. For example, in addition to summary statistics, the data points behind means, medians and variance measures should be available. If there are restrictions on publicly sharing data—e.g. participant privacy or use of data from a third party—those must be specified. 

We are in the process of uploading our FASTA files to the Sequence Read Archive in Genbank, but raw data is also provided in supplemental files.

  Reviewer #1: Dear Editor, the paper “Characterizing changes in soil microbiome abundance and diversity due to different cover crop techniques” is a study of the microbial composition of soils treated with single and multi-mix covercropping throughout a one year activity in organic agriculture regimen. The article structure is sound, being organized around the hypothesis that multi-mix cover crop has an advantage on simpler cover crop choices. The statistics and the microbial methodology are correct. There are some major and minor points to improve, as listed below.  

Major points 1. I have some doubts on the meaning and effect of these tiny experimental plots and if (even in the center of each plot) the border effect can cause problems of any type. In any case a discussion on this point seems necessary as well as the position of the sampling within the plot

We appreciate this reviewer’s comments. We address these points in the methods and discussion sections and included additional references regarding the border effect (lines 803-813). 

  

2. The OUTs are described only by stating that the standard QIME setting was used. In my experience, sometimes, QIME is indeed set up to define ASV rather than OTUs. The great number of OTUs makes me think that indeed they could be ASV groups. In any case, for clarity sake, the authors should state what was the similarity threshold chosen.

QIIME was set up to produce OTUs not ASV. The similarity cutoff in QIIME OTU is 97%. This has been addressed in the Methods section, line 249.

3. As an addition, I would also encourage the authors to briefly describe the variability within the OTUs in order to understand whether they are largely clonal or real OTUs with a species like structure.

Based on the similarity cutoff, OTUS that share ≥ 97% sequence similarity are collapsed into one, and this has been clarified in the methods (line 249). Given that the 16S genes are highly conserved, it is not a surprise that there is a high similarity level among the OTUs. We are concerned that more stringency in the OTU picking could over cluster and lead to a hiding of legitimate variations. Since our analysis is more focused on the higher taxonomical levels (like phyla), we don’t believe we need to be so stringent at the OTU picking level. 

4. The authors claim that the changes of the mixes are induced by variations in the rhizosphere, which is quite likely. Now the point is why did not the authors sample the soil among the roots separately from the rest if they wanted to propose and support this hypothesis? Is there any other indirect evidence in favor of this very likely hypothesis, beyond the likelihood itself?  

Our samples were taken from the rhizosphere when appropriate, however time point one occurred before any planting and thus no rhizosphere was present. We have clarified our sampling technique in the methods section (lines 163-184).

5. The authors state at the beginning that this is one of the few papers on the one-year effect of cover cropping but fail to draw any conclusion about it in the course of the paper. For instance, the relative insensitivity or carbon content to the treatments, can be an effect of the short experimental time?  

We appreciate this critique and have addressed this in the discussion and included relevant references and more discussion on the meaning of our results in terms of the experimental timeline (lines 578-592). 

6. In general the paper should be a bit more condensed to ease its reading. The discussion is partly an enlargement of the result section, rather than a real discussion.

We have attempted to shorten and streamline the discussion to avoid too much of a reiteration of the results.

 Minor points 1. Line 165- it is unclear how can the authors sample 500 ul, i.e half cubic centimeter, from a sampling cylinder with a 16 mm diameter and 100 mm depth. I would conclude that it is 64*pi *100 = 20.096 cubic millimeters i.e. ca 20 mL. In any case the author are encouraged to use Standard measures and therefore mm for lengths an mL or uL (no ul ) for volumes.

We have changed the measurements in the methods as suggested by the reviewer (lines 184-187).

2. Line 199 – Fragment Analyzer. Is it the Agilent tool? If so state it, please, along with the experimental settings.

We used Fragment Analyzer System (Agilent) to assess the libraries size using the high sensitivity NGS kit. We have included this additional information in the methods (line 229).

 

3. Lines 208-209 – rephrase: it is hard to understand

This has been rephrased to improve clarity (now lines 244-246)

 4. Line 218 – is it CRAN R anosim function? If so state it and acknowledge it with a proper citation. Moreover state its settings.

Statistical significance analysis: The “compare_categories.py” function in QIIME was used to analyze the strength and significance of the differences among samples of each group. We use the statistical method ANOSIM, which uses “bray” distance as the default dissimilarity function. The used dissimilarity function is “bray” [ distance = bray]. This is the default function in the ANSIOM R function. We have updated our methods section to reflect these additions (lines 256-275).  

5. ANOVA and KW were carried out with or what?

These statistical tests were carried out with our compositional relative abundance data. A more detailed explanation can be found below in response to Reviewer 2 #5.  

6. Table 1, please separate thousands with the standard “,” in order to make the figures more readable.

Table 1 has been modified with the standard “,”. Page 12 line 350

 7. Line 564 – sentence without the verb, apparently.

This has been resolved (currently now line 702)

8. Iconography taxonomic names not in italic.

This has been resolved.

Reviewer #2: Review  1. I don't believe the authors have submitted raw sequencing data to a public repository, such as NCBI SRA. While it might not be a strict requirement, doing so greatly improves the reproducibility of the study and enables other researchers to improve on the analyses carried out by the authors.

We appreciate this reviewer’s dedication to open access data. We are currently working on uploading the numerous FASTA files to SRA. 

2. (Lines 162-163) The authors should explain, why there is only one replicate per subplot for the first time-point. Considering that the authors have used ANOVA and ANOSIM, both of which are quite sensitive to sample imbalances, the authors should emphasise this issue.

We have clarified this reasoning in the methods section. Ultimately, we chose 9 samples (one for each subplot) at the first timepoint since this was prior to any soil manipulation and thus we did not believe that more samples were necessary or cost effective. 

 

3. (Lines 210-213) OTU picking is an extremely noisy process that is extremely prone to overestimating absolute diversity (i.e. the number of observed OTUs) [1,2,3,4]. Moreover, UCLUST is one of the least accurate heuristic sequence clustering algorithms [3]. The authors must apply denoising to reconstruct exact amplicon sequence variations using DADA2 [4] or Deblur.

The reviewer is right to be concerned about this noise in the OTU-picking process. While the ESV approach is much more stringent and uses only identical and unique 16S sequences in the downstream analysis, it has been shown to lead to 1) losing significant portion of the sequencing data due to its significance to data quality, 2) too much diversity, and hides/conceals the divergence in rRNA operons. This topic is discussed in details here (http://fiererlab.org/2017/05/02/lumping-versus-splitting-is-it-time-for-microbial-ecologists-to-abandon-otus/)

We did a comparison between DADA2 (ESV) and QIIME(OTU) on a portion of our data. Our goal was to assess if using either of the two methods leads to the identification of more/less/different phyla, given that our key results are at the phyla level. Our results (multiple tables and figures) can be found in the uploaded response letter. These data demonstrate that both approaches identify the same types and numbers of the top abundant phyla. Thus, we believe that the use of QIIME is appropriate for our data. We would be happy to include any of this as supplemental information, as deemed appropriate by the editor.

 

4. (Line 218) ANOSIM operates on dissimilarity matrices: the authors must specify the (dis)similarity functions they've used to generate these matrices.

We used dissimilarity function “bray” [ distance = bray]. This is the default function in the ANSIOM R function. This has been updated in more detail in the methods section (lines 256-275).

 5. (Lines 219-221, 229) ANOVA, Kruskal-Wallis and t-test cannot be applied to raw compositional data, such as sequencing data and chemical compositions [5,6]: the authors must carry out statistical analyses designed for compositional data.

We appreciate this critique and the references provided. We would like to refer the reviewer to the following two resources:

1. Hypothesis testing and statistical analysis of microbiome: https://www.ncbi.nlm.nih.gov/pmc/articles/PMC6128532/

2. QIIME: http://qiime.org/scripts/group_significance.html

The above-mentioned statistical methods and their suitability for the microbiome sequencing data are discussed. In addition, these methods are applied on normalized count matrix, which is the same as RNAseq and microarray data at this point. These statistical methods are routinely used for significance analysis between groups, and as such we believe that this is the appropriate statistical approaches for our dataset. We hesitate to add these explanations to our methods section given the present length of the manuscript, however if the editor thinks this would be a valuable addition then we would be happy to add it.

 6. (Lines 271, 283-285) Absolute diversity analysis is neither reliable (due to aforementioned issues with OTU picking), not subcompositionally coherent. The same critique applies to Chao and Shannon indices. If the authors do want to conduct alpha-diversity analysis, they can use the Aitchison's norm. Otherwise, alpha-diversity analyses must be removed. 

To estimate both alpha and beta diversity, we used the QIIME functions alpha_rarefraction.py and beta_diversty.py, respectively. These functions do normalize the data using the rarefaction method. This is described in detail here: 

1. Alpha diversity: http://qiime.org/scripts/alpha_diversity.html

2. Alpha diversity analysis: https://twbattaglia.gitbooks.io/introduction-to-qiime/content/alpha_analysis.html

3. Beta diversity: http://qiime.org/scripts/beta_diversity.html

We made a quick comparison between Rarefaction and Aithchison normalization methods in two time points, please refer the associated tables and plots in the uploaded response letter.

In timepoint 1, the statistical tests show that both rarefaction and Aitchison are different. However, both methods produce similar statistical tests (in terms of significance) in timepoint 2. We therefor think that using the rarefaction normalization prior to alpha and beta diversity estimation is as good as the Aitchison normalization. If the editor deems it appropriate, we are glad to add these explanations to the methods section of the paper or as supplemental data.

---

## [Decision Letter · Decision Letter 1]

14 Jan 2020

PONE-D-19-19916R1

Characterizing changes in soil microbiome abundance and diversity due to different cover crop techniques

PLOS ONE

Dear Dr Fankhauser,

Thank you for submitting your manuscript to PLOS ONE. After careful consideration, we feel that it has merit but does not fully meet PLOS ONE’s publication criteria as it currently stands. Therefore, we invite you to submit a revised version of the manuscript that addresses the points raised during the review process.

We would appreciate receiving your revised manuscript by Feb 28 2020 11:59PM. To enhance the reproducibility of your results, we recommend that if applicable you deposit your laboratory protocols in protocols.io, where a protocol can be assigned its own identifier (DOI) such that it can be cited independently in the future. For instructions see: http://journals.plos.org/plosone/s/submission-guidelines#loc-laboratory-protocols

We look forward to receiving your revised manuscript.

Kind regards,

Primo Proietti

Academic Editor

PLOS ONE

Additional Editor Comments (if provided):

Please, based on the comments from Reviewer 2, try to improve the manuscript. Thank you

Reviewers' comments:

Reviewer's Responses to Questions

**Comments to the Author**

1. If the authors have adequately addressed your comments raised in a previous round of review and you feel that this manuscript is now acceptable for publication, you may indicate that here to bypass the “Comments to the Author” section, enter your conflict of interest statement in the “Confidential to Editor” section, and submit your "Accept" recommendation.

Reviewer #1: All comments have been addressed

Reviewer #2: All comments have been addressed

2. Is the manuscript technically sound, and do the data support the conclusions?

Reviewer #1: Yes

Reviewer #2: Partly

3. Has the statistical analysis been performed appropriately and rigorously? 

Reviewer #1: Yes

Reviewer #2: No

4. Have the authors made all data underlying the findings in their manuscript fully available?

Reviewer #1: Yes

Reviewer #2: No

5. Is the manuscript presented in an intelligible fashion and written in standard English?

Reviewer #1: Yes

Reviewer #2: Yes

6. Review Comments to the Author

Reviewer #1: The authors amended quite well their paper and gave exhaustive and satisfactory answers to the question posed.

Reviewer #2: 1. We appreciate this reviewer’s dedication to open access data. We are currently working on uploading the numerous FASTA files to SRA.

Thank you very much for this decision. I am looking forward to seeing your data uploaded to SRA.

2. We have clarified this reasoning in the methods section. Ultimately, we chose 9 samples (one for each subplot) at the first timepoint since this was prior to any soil manipulation and thus we did not believe that more samples were necessary or cost effective.

I believe, you refer to the following sentence (lines 159-161): "There was only 1 sample per subplot taken at time point 1 since the soil had just been evenly tilled across each subplot and no further manipulation had been undertaken; thus the extra replicates of the plots were deemed unnecessary".

Even if that's the case, you should have considered the effect sample imbalance has on statistical methods you've chosen. ANOSIM, in particular, is extremely sensitive to sample size imbalance and heteroscedasticity (that accompanies sample imbalances) [1]. If you are going to keep ANOSIM in your analysis, you should leave an appropriate warning about sample imbalances.

3. The reviewer is right to be concerned about this noise in the OTU-picking process. While the ESV approach is much more stringent and uses only identical and unique 16S sequences in the downstream analysis, it has been shown to lead to 1) losing significant portion of the sequencing data due to its significance to data quality, 2) too much diversity, and hides/conceals the divergence in rRNA operons. This topic is discussed in details here (http://fiererlab.org/2017/05/02/lumping-versus-splitting-is-it- time-for-microbial-ecologists-to-abandon-otus/). We did a comparison between DADA2 (ESV) and QIIME(OTU) on a portion of our data. Our goal was to assess if using either of the two methods leads to the identification of more/less/different phyla, given that our key results are at the phyla level. Our results (multiple tables and figures) can be found in the uploaded response letter. These data demonstrate that both approaches identify the same types and numbers of the top abundant phyla. Thus, we believe that the use of QIIME is appropriate for our data. We would be happy to include any of this as supplemental information, as deemed appropriate by the editor.

I have given multiple peer-reviewed references showing how bad OTU-picking, (and UCLUST-based OTU-picking in particular) is at recovering microbial diversity from amplicon sequences. Considering that you are working with the total number of OTUs in your analysis, the fact that OTU-picking has been shown to dramatically (from 10^1 to 10^3 times) inflate this number is absolutely damning, even if the procedure produces adequate composition at the level of phyla. With all due respect, a single unreviewed blog-post is not a valid source of evidence. Moreover, the very people behind QIIME's original preprocessing pipeline implore their users to abandon it in favour of denoising: "... This process, also known as OTU picking, was once a common procedure, used to simultaneously dereplicate but also perform a sort of quick-and-dirty denoising procedure (to capture stochastic sequencing and PCR errors, which should be rare and similar to more abundant centroid sequences). Use denoising methods instead if you can. Times have changed. Welcome to the future" [ https://docs.qiime2.org/2019.10/tutorials/overview/ ].

4. ANOSIM operates on dissimilarity matrices: the authors must specify the (dis)similarity functions they've used to generate these matrices.

Bray-Curtis is not subcompositionally coherent – you should add an appropriate warning. You might consider adding another ANOSIM assay based on Aitchison distances.

5. We appreciate this critique and the references provided. We would like to refer the reviewer to the following two resources... In addition, these methods are applied on normalized count matrix, which is the same as RNAseq and microarray data at this point. These statistical methods are routinely used for significance analysis between groups, and as such we believe that this is the appropriate statistical approaches for our dataset.

I am not sure you have paid due attention to my references, but I will return to them later. As for the references you have provided, the second one is a link to QIIME1 documentation. Since QIIME1 is a woefully outdated piece of software that has been discouraged by its own authors, its documentation doesn't look like a valid reference to rely on. That leaves the first reference, i.e. "Hypothesis testing and statistical analysis of microbiome" by Xia and Sun. First of all, this paper is a review of methods that have been used in microbiome research regardless of statistical correctness. Second of all, the very paper contains the following sentences:

"First, the existing statistical methods for analyzing microbiome proportional data do not solve constraint problem, and some researchers even do not know it exists. Most standard statistical methods, such as the Pearson correlation coefficient, t-test, ANOVA are still widely used or exist in current literature on the analysis of microbiome data[23, 24, 25, 26] without testing the data distribution and transformation. One assumption of standard statistical methods is that the compared data are independent. Since the sum of the relative abundances is unity, it indicates the data are not independent with the unity or any constant constrain. Thus it is not appropriate to directly use these methods for analyzing microbiome relative abundance data".

In other words, your own reference reinforces my line of criticism. The fact that you are using a "normalized count matrix, which is the same as RNAseq" changes nothing, because no normalisation procedure can remove the total-sum-constraint problem (it can only change the total sum). Furthermore, RNAseq methods operate under strong assumptions (i.e. there must be many transcripts, but few of them can vary across states) that simply do not apply to microbiome data (whether these assumptions hold in RNAseq studies is a separate issue). Even ALDEx2 is not ideal for microbiome research, because it simply applies the CLR transform (which is similar to the normalisation procedure from DESeq2) and operates under the same assumptions as other RNAseq packages. This is, of course, covered in depth in the references I have given in the previous response.

Here I would like to reiterate a line from my previous response: the authors must carry out statistical analyses designed for compositional data.

6. We made a quick comparison between Rarefaction and Aithchison normalization.

It is a norm, not a normalisation procedure. How have you treated zeros?

References

1. Marti Anderson and Daniel Walsh: PERMANOVA, ANOSIM, and the Mantel test in the face of heterogeneous dispersions: What null hypothesis are you testing? – Ecological Monographs, Vol. 83, No. 4 (November 2013), pp. 557-574.

7. PLOS authors have the option to publish the peer review history of their article (what does this mean?). If published, this will include your full peer review and any attached files.

Reviewer #1: Yes: Gianluigi Cardinali

Reviewer #2: No

---

## [Author Response · Author response to Decision Letter 1]

20 Mar 2020

Reviewer #1: The authors amended quite well their paper and gave exhaustive and satisfactory answers to the question posed.

Reviewer #2: 1. We appreciate this reviewer’s dedication to open access data. We are currently working on uploading the numerous FASTA files to SRA.

Thank you very much for this decision. I am looking forward to seeing your data uploaded to SRA.

Given the reanalysis of our data, and the new files, we are still in the process of uploading our fastq data. We have also included substantial amount of supplemental data in our submission.

2. I believe, you refer to the following sentence (lines 159-161): "There was only 1 sample per subplot taken at time point 1 since the soil had just been evenly tilled across each subplot and no further manipulation had been undertaken; thus the extra replicates of the plots were deemed unnecessary".

Even if that's the case, you should have considered the effect sample imbalance has on statistical methods you've chosen. ANOSIM, in particular, is extremely sensitive to sample size imbalance and heteroscedasticity (that accompanies sample imbalances) [1]. If you are going to keep ANOSIM in your analysis, you should leave an appropriate warning about sample imbalances.

We have noted the limitation of this analysis in the methods, lines 160-163.

3. I have given multiple peer-reviewed references showing how bad OTU-picking, (and UCLUST-based OTU-picking in particular) is at recovering microbial diversity from amplicon sequences. Considering that you are working with the total number of OTUs in your analysis, the fact that OTU-picking has been shown to dramatically (from 10^1 to 10^3 times) inflate this number is absolutely damning, even if the procedure produces adequate composition at the level of phyla. With all due respect, a single unreviewed blog-post is not a valid source of evidence. Moreover, the very people behind QIIME's original preprocessing pipeline implore their users to abandon it in favour of denoising: "... This process, also known as OTU picking, was once a common procedure, used to simultaneously dereplicate but also perform a sort of quick-and-dirty denoising procedure (to capture stochastic sequencing and PCR errors, which should be rare and similar to more abundant centroid sequences). Use denoising methods instead if you can. Times have changed. Welcome to the future" [ https://docs.qiime2.org/2019.10/tutorials/overview/ ].

We thank the reviewer in their persistence with this, and we have heeded the advice. We have completely reanalyzed the data using QIIME 2 and performed a denoising step (described in the methods lines 219-221 and in the results lines 287 and 301 ). This is further represented in Table 1. We believe that this reanalysis has revealed new and interesting trends which we discuss extensively in our results and discussion sections. 

4. Bray-Curtis is not subcompositionally coherent – you should add an appropriate warning. You might consider adding another ANOSIM assay based on Aitchison distances.

We have performed an ANOSIM on the Aitchison distances, the Aitchison distance values are included in supplemental tables 2 and 3 (S2 Table, S3 Table), and the resulting processed data was analyzed using multivariate comparative methods such as ANOVA and ANOSIM, the results were reported in results Tables 2A and 2B.

5. I am not sure you have paid due attention to my references, but I will return to them later. As for the references you have provided, the second one is a link to QIIME1 documentation. Since QIIME1 is a woefully outdated piece of software that has been discouraged by its own authors, its documentation doesn't look like a valid reference to rely on. That leaves the first reference, i.e. "Hypothesis testing and statistical analysis of microbiome" by Xia and Sun. First of all, this paper is a review of methods that have been used in microbiome research regardless of statistical correctness. Second of all, the very paper contains the following sentences:

"First, the existing statistical methods for analyzing microbiome proportional data do not solve constraint problem, and some researchers even do not know it exists. Most standard statistical methods, such as the Pearson correlation coefficient, t-test, ANOVA are still widely used or exist in current literature on the analysis of microbiome data[23, 24, 25, 26] without testing the data distribution and transformation. One assumption of standard statistical methods is that the compared data are independent. Since the sum of the relative abundances is unity, it indicates the data are not independent with the unity or any constant constrain. Thus it is not appropriate to directly use these methods for analyzing microbiome relative abundance data".

In other words, your own reference reinforces my line of criticism. The fact that you are using a "normalized count matrix, which is the same as RNAseq" changes nothing, because no normalisation procedure can remove the total-sum-constraint problem (it can only change the total sum). Furthermore, RNAseq methods operate under strong assumptions (i.e. there must be many transcripts, but few of them can vary across states) that simply do not apply to microbiome data (whether these assumptions hold in RNAseq studies is a separate issue). Even ALDEx2 is not ideal for microbiome research, because it simply applies the CLR transform (which is similar to the normalisation procedure from DESeq2) and operates under the same assumptions as other RNAseq packages. This is, of course, covered in depth in the references I have given in the previous response.

Here I would like to reiterate a line from my previous response: the authors must carry out statistical analyses designed for compositional data.

We have conduced statistical analyses designed for compositional data by transforming our data to Aitchison’s distances. We used the ANOSIM and ANOVA method to test whether the treatment among 3 groups are significantly different. According to the reviewer 2’s suggestion, the Aitchison’s distance is applied to calculate the distance matrix (the documentation is here: https://library.qiime2.org/plugins/deicode/19/ ). This is described in both the methods (starting on line 224) and results sections (starting on line 319).

6. It is a norm, not a normalisation procedure. How have you treated zeros?

We have described our analysis, using Aitchison norm, starting on line 224 (statistical anlaysis) in the methods. We provide additional details on our use of Aitchison norm in the results section. We used the cutoff at 1 for each feature, so it will ignore the features with 0 count.

---

## [Editor Report · Decision Letter 2]

13 Apr 2020

PONE-D-19-19916R2

Characterizing changes in soil microbiome abundance and diversity due to different cover crop techniques

PLOS ONE

Dear Dr Fankhauser,

Thank you for submitting your manuscript to PLOS ONE. After careful consideration, we feel that it has merit but does not fully meet PLOS ONE’s publication criteria as it currently stands. Therefore, we invite you to submit a revised version of the manuscript that addresses the points raised during the review process.

We would appreciate receiving your revised manuscript by May 28 2020 11:59PM. To enhance the reproducibility of your results, we recommend that if applicable you deposit your laboratory protocols in protocols.io, where a protocol can be assigned its own identifier (DOI) such that it can be cited independently in the future. For instructions see: http://journals.plos.org/plosone/s/submission-guidelines#loc-laboratory-protocols

We look forward to receiving your revised manuscript.

Kind regards,

Primo Proietti

Academic Editor

PLOS ONE

Additional Editor Comments (if provided):

Dear Author,

considering the comments of reviewer 2, your intervention on the manuscript is still necessary.

Cordiality

---

## [Author Response · Author response to Decision Letter 2]

14 Apr 2020

Below are the responses originally submitted with our resubmission on March 20, 2020:

Reviewer #1: The authors amended quite well their paper and gave exhaustive and satisfactory answers to the question posed.

Reviewer #2: 1. We appreciate this reviewer’s dedication to open access data. We are currently working on uploading the numerous FASTA files to SRA.

Thank you very much for this decision. I am looking forward to seeing your data uploaded to SRA.

Given the reanalysis of our data, and the new files, we are still in the process of uploading our fastq data. We have also included substantial amount of supplemental data in our submission.

2. I believe, you refer to the following sentence (lines 159-161): "There was only 1 sample per subplot taken at time point 1 since the soil had just been evenly tilled across each subplot and no further manipulation had been undertaken; thus the extra replicates of the plots were deemed unnecessary".

Even if that's the case, you should have considered the effect sample imbalance has on statistical methods you've chosen. ANOSIM, in particular, is extremely sensitive to sample size imbalance and heteroscedasticity (that accompanies sample imbalances) [1]. If you are going to keep ANOSIM in your analysis, you should leave an appropriate warning about sample imbalances.

We have noted the limitation of this analysis in the methods, lines 160-163.

3. I have given multiple peer-reviewed references showing how bad OTU-picking, (and UCLUST-based OTU-picking in particular) is at recovering microbial diversity from amplicon sequences. Considering that you are working with the total number of OTUs in your analysis, the fact that OTU-picking has been shown to dramatically (from 10^1 to 10^3 times) inflate this number is absolutely damning, even if the procedure produces adequate composition at the level of phyla. With all due respect, a single unreviewed blog-post is not a valid source of evidence. Moreover, the very people behind QIIME's original preprocessing pipeline implore their users to abandon it in favour of denoising: "... This process, also known as OTU picking, was once a common procedure, used to simultaneously dereplicate but also perform a sort of quick-and-dirty denoising procedure (to capture stochastic sequencing and PCR errors, which should be rare and similar to more abundant centroid sequences). Use denoising methods instead if you can. Times have changed. Welcome to the future" [ https://docs.qiime2.org/2019.10/tutorials/overview/ ].

We thank the reviewer in their persistence with this, and we have heeded the advice. We have completely reanalyzed the data using QIIME 2 and performed a denoising step (described in the methods lines 219-221 and in the results lines 287 and 301 ). This is further represented in Table 1. We believe that this reanalysis has revealed new and interesting trends which we discuss extensively in our results and discussion sections. 

4. Bray-Curtis is not subcompositionally coherent – you should add an appropriate warning. You might consider adding another ANOSIM assay based on Aitchison distances.

We have performed an ANOSIM on the Aitchison distances, the Aitchison distance values are included in supplemental tables 2 and 3 (S2 Table, S3 Table), and the resulting processed data was analyzed using multivariate comparative methods such as ANOVA and ANOSIM, the results were reported in results Tables 2A and 2B.

5. I am not sure you have paid due attention to my references, but I will return to them later. As for the references you have provided, the second one is a link to QIIME1 documentation. Since QIIME1 is a woefully outdated piece of software that has been discouraged by its own authors, its documentation doesn't look like a valid reference to rely on. That leaves the first reference, i.e. "Hypothesis testing and statistical analysis of microbiome" by Xia and Sun. First of all, this paper is a review of methods that have been used in microbiome research regardless of statistical correctness. Second of all, the very paper contains the following sentences:

"First, the existing statistical methods for analyzing microbiome proportional data do not solve constraint problem, and some researchers even do not know it exists. Most standard statistical methods, such as the Pearson correlation coefficient, t-test, ANOVA are still widely used or exist in current literature on the analysis of microbiome data[23, 24, 25, 26] without testing the data distribution and transformation. One assumption of standard statistical methods is that the compared data are independent. Since the sum of the relative abundances is unity, it indicates the data are not independent with the unity or any constant constrain. Thus it is not appropriate to directly use these methods for analyzing microbiome relative abundance data".

In other words, your own reference reinforces my line of criticism. The fact that you are using a "normalized count matrix, which is the same as RNAseq" changes nothing, because no normalisation procedure can remove the total-sum-constraint problem (it can only change the total sum). Furthermore, RNAseq methods operate under strong assumptions (i.e. there must be many transcripts, but few of them can vary across states) that simply do not apply to microbiome data (whether these assumptions hold in RNAseq studies is a separate issue). Even ALDEx2 is not ideal for microbiome research, because it simply applies the CLR transform (which is similar to the normalisation procedure from DESeq2) and operates under the same assumptions as other RNAseq packages. This is, of course, covered in depth in the references I have given in the previous response.

Here I would like to reiterate a line from my previous response: the authors must carry out statistical analyses designed for compositional data.

We have conduced statistical analyses designed for compositional data by transforming our data to Aitchison’s distances. We used the ANOSIM and ANOVA method to test whether the treatment among 3 groups are significantly different. According to the reviewer 2’s suggestion, the Aitchison’s distance is applied to calculate the distance matrix (the documentation is here: https://library.qiime2.org/plugins/deicode/19/ ). This is described in both the methods (starting on line 224) and results sections (starting on line 319).

6. It is a norm, not a normalisation procedure. How have you treated zeros?

We have described our analysis, using Aitchison norm, starting on line 224 (statistical anlaysis) in the methods. We provide additional details on our use of Aitchison norm in the results section. We used the cutoff at 1 for each feature, so it will ignore the features with 0 count.

---

## [Editor Report · Decision Letter 3]

16 Apr 2020

Characterizing changes in soil microbiome abundance and diversity due to different cover crop techniques

PONE-D-19-19916R3

Dear Dr. Fankhauser,

We are pleased to inform you that your manuscript has been judged scientifically suitable for publication and will be formally accepted for publication once it complies with all outstanding technical requirements.

With kind regards,

Primo Proietti

Academic Editor

PLOS ONE

Additional Editor Comments (optional):

The authors substantially improved the manuscript by taking into account the reviewers' requests/comments. In its current form, it can be published in Plos One

All the best
---

## [Editor Report · Acceptance letter]

23 Apr 2020

PONE-D-19-19916R3 

Characterizing changes in soil microbiome abundance and diversity due to different cover crop techniques 

Dear Dr. Fankhauser:

I am pleased to inform you that your manuscript has been deemed suitable for publication in PLOS ONE. Congratulations! Your manuscript is now with our production department. 

With kind regards,

on behalf of

Dr. Primo Proietti 

Academic Editor

PLOS ONE